# Closing the Modality Gap for Mixed Modality Search

## Abstract

Mixed modality search, retrieving information across a heterogeneous corpus composed of images, texts, and multimodal documents, is an important yet underexplored real-world application. In this work, we investigate how contrastive vision-language models, such as CLIP, perform on the mixed modality search task. Our analysis reveals a critical limitation: these models exhibit a pronounced modality gap in the embedding space, where image and text embeddings form distinct clusters, leading to intra-modal ranking bias and inter-modal fusion failure. To address this issue, we propose GR-CLIP, a lightweight post-hoc calibration method that removes the modality gap in CLIP's embedding space. Evaluated on MixBench, the first benchmark specifically designed for mixed modality search, GR-CLIP improves NDCG@10 by up to 26% over CLIP, surpasses recent vision-language generative embedding models by 4%, while using $75\times$ less compute.

## 1 Introduction

Information in the digital world exists across multiple modalities—text, images, video, audio, and their various combinations. While traditional retrieval systems have primarily focused on searching within a **homogeneous** corpus (e.g., text-to-text or text-to-image retrieval) (Robertson et al., 2009; Karpukhin et al., 2020; Lee et al., 2018; Radford et al., 2021), real-world applications increasingly demand the ability to search and retrieve relevant content across **heterogeneous** modalities (e.g., text-to-{text, image, or both} retrieval) (Voyage AI, 2024). For instance, a user searching for "Mountain Fuji" might expect to find text documents, standalone images, and multimodal webpages that combine both modalities to describe the mountain (Figure 1a).

Despite its practical importance, the task of **mixed modality search** remains largely underexplored (Voyage AI, 2024). The central challenge lies in constructing a unified embedding space where semantically similar content across modalities—such as an image and a textual description of "Mountain Fuji"—can be mapped to nearby locations. This enables accurate measurement of semantic similarity between queries and documents, regardless of their modality. Recent advances in multimodal contrastive learning, particularly CLIP-based models (Radford et al., 2021; Cherti et al., 2023; Zhai et al., 2023), offer a promising solution by aligning text and image embeddings through training on large-scale paired image-text datasets.

In this work, we investigate how well these contrastive models perform in realistic mixed modality search scenarios. Specifically, CLIP consists of two separate encoders for vision and language (Radford et al., 2021). For each corpus item, we encode image-only and text-only documents using their respective encoders. For multimodal documents containing both image and text, we compute a linear combination of the image and text embeddings to represent them (Figure 1b). Once the embeddings are obtained, we perform similarity search by computing cosine similarity between the query embedding and each corpus item, and evaluate performance using standard retrieval metrics such as NDCG@10 (Järvelin & Kekäläinen, 2002), which measures the quality of the top-10 ranked results based on relevance.

Our analysis reveals a fundamental limitation of CLIP-style contrastive models: they exhibit a pronounced **modality gap** (Liang et al., 2022; Zhang et al., 2023; 2024) in their embedding space, significantly degrading retrieval performance in mixed modality settings. Although these models are trained to align image-text pairs, image and text embeddings form separate clusters and remain far apart in the embedding space (Figure 1c). This clustering causes a strong *intra-modal ranking bias*

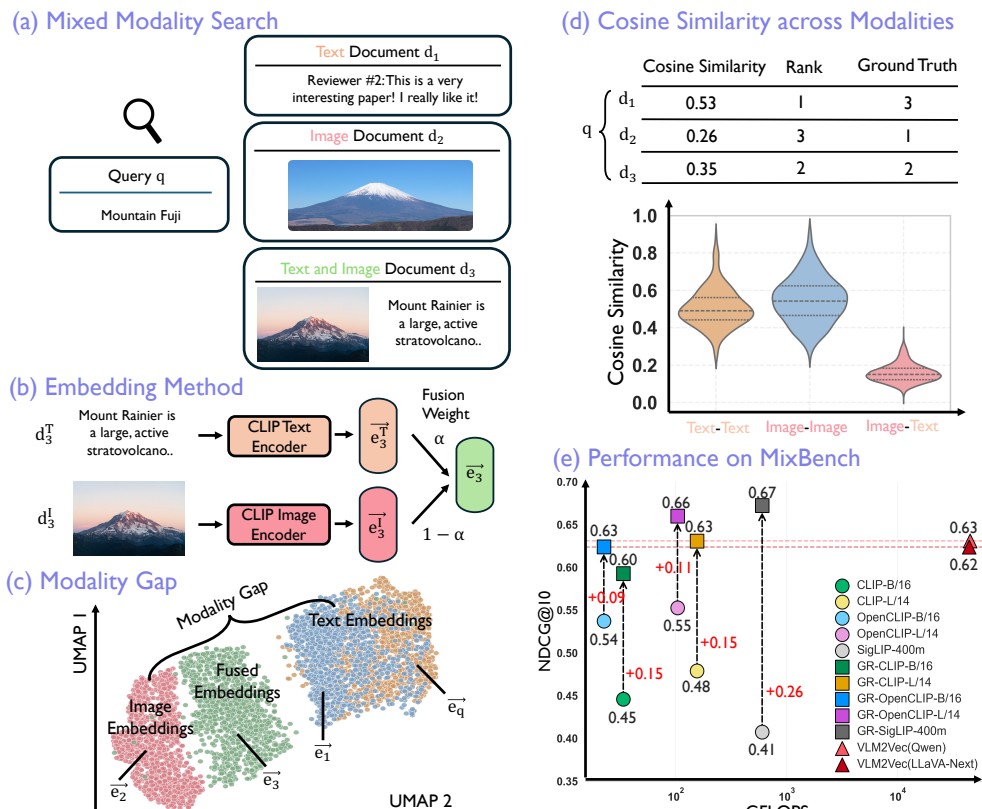

Figure 1: **Overview of mixed modality search. (a) Problem Formulation:** Mixed modality search aims to retrieve relevant information from a heterogeneous corpus containing multimodal documents. This is achieved by embedding both the query and documents, followed by similarity-based retrieval. **(b) Embedding Method:** Unimodal documents are embedded using CLIP's modality-specific encoder, while multimodal documents are embedded via a weighted fusion of image and text features. **(c) Modality Gap:** CLIP's embedding space exhibits a modality gap: embeddings form distinct clusters for each modality and remain largely separated across modalities. **(d) Cosine Similarity Across Modalities:** Due to this modality gap, documents that share the same modality as the query tend to have higher cosine similarity scores and are ranked higher, introducing systematic ranking bias. **(e) Performance on MixBench:** On our newly created MixBench benchmark—specifically designed for the task of mixed modality search—GR-CLIP, a lightweight post-hoc calibration method that closes the modality gap, significantly improves performance and outperforms the state-of-the-art VLM2Vec (Jiang et al., 2025b) baseline with substantially lower computational cost.

(§3), where similarities between items of the same modality (e.g., image-to-image or text-to-text) are much higher than those across modalities (e.g., image-to-text), skewing retrieval rankings (Figure 1d). For instance, given the text query "Mountain Fuji," an image that depicts Mountain Fuji is ranked below an unrelated text like "a great paper." Additionally, the modality gap hurts *inter-modal fusion* (§4): combining image and text embeddings via linear interpolation often pushes the features to a suboptimal region, weakening their semantics and performing worse than using image or text alone.

To address the ranking bias and fusion failure caused by the modality gap, we introduce **GR-CLIP**, a lightweight post-hoc calibration method that removes the modality gap in CLIP's embedding space (GR stands for gap-removed). Prior work (Zhang et al., 2023; 2024) has shown that the modality gap in CLIP-like models can be approximated by a constant vector that is orthogonal to the image and text embedding subspaces. Building on the mean-shift formulation proposed in (Zhang et al., 2024), we compute the mean embeddings of all image and text data, use their difference to estimate the modality gap, and subtract this vector from all embeddings before performing retrieval. This method requires only a single pass over the dataset to compute mean embeddings and introduces negligible computational overhead.

Evaluated on **MixBench**, our benchmark explicitly designed for mixed modality search with four subsets (Google-WIT (Srinivasan et al., 2021), MSCOCO (Lin et al., 2014), OVEN (Hu et al., 2023), VisualNews (Liu et al., 2021)), GR-CLIP consistently outperforms the original CLIP models, achieving up to 26 percentage points improvement in NDCG@10. It also surpasses recent vision-language generative embedding methods such as VLM2Vec (Jiang et al., 2025b) by 4 percentage points, while reducing computational cost by $75\times$. Furthermore, we demonstrate that our method generalizes across different CLIP variants (e.g., OpenAI CLIP (Radford et al., 2021), OpenCLIP (Cherti et al., 2023), SigLIP (Zhai et al., 2023)) and modalities (e.g., text-to-image, text-to-audio, text-to-video).

In summary, our contribution lies in problem identification, benchmark creation, and empirical analysis. We are the first to formulate and study the problem of **mixed modality search**, which reflects real-world scenarios such as web search engines, where users query a heterogeneous corpus containing diverse modality types. We demonstrate that state-of-the-art contrastive models suffer from ranking bias and fusion failure due to the modality gap, and we adapt a lightweight post-hoc calibration method to address this issue. Our findings highlight the importance of constructing truly unified embedding spaces for effective mixed modality search.

## 2 PRELIMINARIES

In this section, we define the mixed modality search task, introduce its challenges and three settings related to the challenge, and describe the methods and evaluation metrics used.

### 2.1 PROBLEM FORMULATION

Mixed modality search aims to retrieve semantically relevant content when both the query and the documents may consist of different combinations of modalities, such as text, image, audio, or video. Let $\mathcal{M}$ be the set of supported modalities (e.g., $\mathcal{M} = \{\text{text, image, audio, video}\}$). A query is denoted by $q$, with modality set $m_q \subseteq \mathcal{M}$. The retrieval corpus is defined as $\mathcal{C} = \{d_i\}_{i=1}^N$, where each document $d_i$ is associated with a modality set $m_i \subseteq \mathcal{M}$. The goal is to compute a similarity score $s(q, d_i)$ for each document and return a ranked list based on semantic relevance, regardless of how the modalities are distributed across queries and documents.

Two properties distinguish mixed modality search from traditional retrieval tasks: **1) heterogeneous corpus:** the modality composition varies across documents, i.e., there exist $d_i, d_j \in \mathcal{C}$ such that $m_i \neq m_j$. For example, one document may be text-only ($m_i = \{\text{text}\}$), another image-only ($m_j = \{\text{image}\}$), and another multimodal ($m_k = \{\text{text, image}\}$); **b) multimodal documents:** some documents contain multiple modalities within a single entry, i.e., $|m_i| > 1$. These modalities often provide complementary information that must be fused for effective understanding (e.g., an image paired with a descriptive caption).

### 2.2 SETTINGS

The combination of a heterogeneous corpus and multimodal documents introduces two central modeling challenges: **1) cross-modal alignment:** ensuring that representations of similar concepts are comparable across different modalities—for instance, the text and image of "Mount Fuji" should be embedded in nearby locations in the representation space; **2) multimodal fusion:** effectively combining multiple modalities within a document to form a unified, semantically meaningful representation—for example, integrating the text and image of "Mount Fuji" to produce a richer representation of the concept. To study these challenges systematically, we define three settings:

**Ablated setting 1: only heterogeneous corpus (§3).** Each document is unimodal ($|m_i| = 1$), but the corpus spans multiple modalities ($|\mathcal{M}| > 1$). For example, it may include text-only and image-only descriptions of the same concept, corresponding to $d_1$ and $d_2$ in Figure 1a. This tests only cross-modal alignment—whether the model can encode comparable representations across modalities.

**Ablated setting 2: only multimodal documents (§4).** All documents contain the same set of modalities ($m_i = \mathcal{M}$ with $|m_i| > 1$). For example, each document includes both an image and a corresponding caption, corresponding to $d_3$ in Figure 1a. This setting focuses purely on multimodal fusion—evaluating whether the model can effectively combine multiple modalities.

**Full setting: mixed modality search (§5).** Documents are variably unimodal or multimodal ($|m_i| \geq 1$), and the corpus is heterogeneous. For instance, some documents may be text-only, others

image-only, and others a combination—corresponding to $d_1$, $d_2$, and $d_3$ all being present in Figure 1a. This is the most realistic and general setting, reflecting real-world corpora such as news articles, product listings, or scientific datasets. It combines both core challenges and serves as our primary evaluation scenario.

## 2.3 METHODS

Given a query $q$ and a document $d_i$, we use an embedding model $f$ to compute their embeddings $e_q = f(q)$ and $e_i = f(d_i)$, and rank documents using cosine similarity: $s(q, d_i) = \frac{e_q \cdot e_i}{\|e_q\| \cdot \|e_i\|}$. We evaluate the following embedding approaches:

**CLIP (baseline) (Radford et al., 2021).** CLIP is a contrastive vision-language model trained to align paired image-text inputs. It encodes each modality separately using an image encoder $f^I$ and a text encoder $f^T$. For unimodal text or image documents $d_i$ and $d_j$, we use the modality-specific encoder to compute the embedding: $e_i = f^I(d_i)$ and $e_j = f^T(d_j)$. For multimodal documents $d_k$ with image and text inputs $d_k^I$ and $d_k^T$, we compute a weighted interpolation: $e_k = \alpha \cdot f^T(d_k^T) + (1 - \alpha) \cdot f^I(d_k^I)$, where $\alpha \in [0, 1]$ balances the contribution of each modality.

**VLM2Vec (baseline) (Jiang et al., 2025b).** VLM2Vec is a state-of-the-art multimodal generative embedding method that adapts large vision-language models $f$ (e.g., LLaVA (Liu et al., 2023a), Qwen-VL (Bai et al., 2023)) to generate document embeddings in an auto-regressive fashion. Each document $d_i$ is formatted as an instruction-style prompt $p_i$ that combines text and image inputs (e.g., *"Generate the embedding for the document: [image tokens] [text tokens]"*), which is then processed autoregressively. The pooled representation from the final decoder layer is used as the embedding $e_i = f(p_i)$. This method captures high-level semantic alignment through joint modeling of the two modalities and instruction tuning.

**GR-CLIP (ours).** Despite CLIP's goal of aligning modalities, prior work reveals a persistent modality gap in its embedding space: image and text embeddings form separate clusters and remain distant (Liang et al., 2022). Given a paired image-text embedding $e_i^T$ and $e_i^I$, their relationship can be modeled as $e_i^T - e_i^I \approx c_\perp$, where $c_\perp$ is a constant vector orthogonal to the shared embedding subspace, representing the modality gap (Zhang et al., 2024). GR-CLIP (GR stands for gap-removed) is a lightweight post-hoc calibration method, adapted from the mean-shift formulation in prior work (Zhang et al., 2024), that removes this gap by subtracting modality-specific means: $e_i'^T = e_i^T - \mathbb{E}_i[e_i^T], e_i'^I = e_i^I - \mathbb{E}_i[e_i^I]$. This zero-centering eliminates the modality gap (Zhang et al., 2024), as $e_i'^T - e_i'^I = (e_i^T - e_i^I) - (\mathbb{E}_i[e_i^T] - \mathbb{E}_i[e_i^I]) \approx c_\perp - c_\perp = 0$, which improves cross-modal alignment at negligible inference cost. For multimodal documents, we apply the same interpolation over the calibrated embeddings. Figure 1b illustrates this process. In practice, we find this simple calibration significantly boosts CLIP's performance and even outperforms VLM2Vec while using much less compute. Additional details about GR-CLIP is in Appendix C.

## 2.4 EVALUATION METRICS

We evaluate retrieval performance using **NDCG@10** (Normalized Discounted Cumulative Gain (Järvelin & Kekäläinen, 2002)), a widely used metric that reflects both the relevance and ranking of the top-10 retrieved documents. Higher NDCG@10 scores indicate better performance. We also report additional metrics in Appendix A.

## 3 RETRIEVAL WITH HETEROGENEOUS CORPUS

As discussed in §2, we begin with an ablated setting of mixed modality search: a heterogeneous corpus composed of unimodal documents (e.g., text-only or image-only; see Figure 2a). This setting evaluates whether a retrieval model can effectively handle the challenge of cross-modal alignment.

### 3.1 DATASET CONSTRUCTION

Since no existing dataset follows this setting, we construct new datasets tailored for this task using two complementary strategies: one based on synthetic screenshots and another using image replacements.

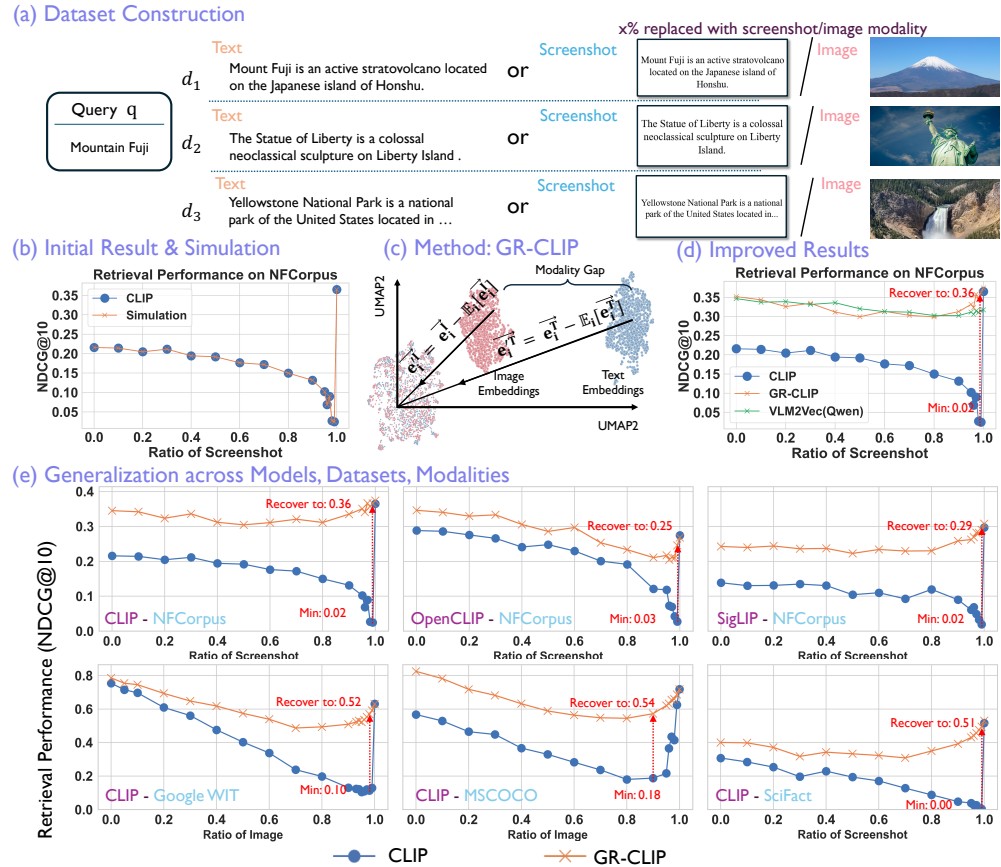

Figure 2: **Retrieval with a heterogeneous corpus. (a) Dataset Construction:** We construct a heterogeneous corpus by randomly replacing text documents with either screenshot renderings of the text or paired images with probability $p$. Since the semantic content remains unchanged, a retrieval system with perfect cross-modal alignment should maintain the same performance regardless of $p$. **(b) Initial Results & Simulation:** Surprisingly, CLIP exhibits a U-shaped performance curve as text is replaced with screenshots. We attribute this behavior to the modality gap in CLIP's embedding space. A simulation experiment that artificially penalizes cross-modal documents reproduces the same U-shaped trend, confirming our hypothesis. **(c) Method — GR-CLIP:** Building on prior work, we propose **GR-CLIP**, a simple post-hoc calibration that removes the modality gap via mean-centering of text and image embeddings. **(d) Improved Results:** GR-CLIP flattens the U-shaped curve and significantly improves retrieval accuracy, achieving comparable or better performance than the VLM2Vec baseline with far less compute. **(e) Generalization Across Models, Datasets, and Modalities:** To evaluate generalization, we test GR-CLIP across three CLIP variants, three additional datasets, and three other modalities (detailed in the Appendix). In all cases, the findings and improvements hold consistently.

**Screenshot replacement.** Starting from a standard text-only retrieval dataset—where both queries and corpus documents are textual—we synthetically render the text documents as image-based screenshots. Specifically, for each text document $d_i^T$, we generate a screenshot version $d_i^I$ containing identical content and replace it with probability $p$ (Figure 2a). This synthetic setup preserves semantic content exactly, making it ideal for controlled experiments. A model with perfect cross-modal alignment should represent paired text and screenshot documents similarly in the embedding space, and thus its retrieval performance should remain unchanged across varying values of $p$. We apply this transformation to two datasets: NFCorpus (Boteva et al., 2016) and SciFact (Wadden et al., 2020) .

**Real image replacement.** For datasets containing image-caption pairs, we replace text captions $d_i^T$ with their corresponding images $d_i^I$ with probability $p$. While this setting is more realistic, it introduces slight semantic differences between modalities. Nevertheless, given the underlying semantic alignment, retrieval performance is expected to remain stable across different replacement

ratios $p$. We construct two datasets using this approach: Google WIT (Srinivasan et al., 2021), MSCOCO (Lin et al., 2014).

## 3.2 INITIAL RESULTS & SIMULATION

We first focus on the synthetic screenshot-based setting due to its exact semantic preservation. Ideally, a model with perfect cross-modal alignment should yield consistent retrieval performance regardless of how many documents are replaced with screenshots.

**Models exhibit a U-shaped performance curve when mixing texts and screenshots.** Surprisingly, we observe a U-shaped performance curve (Figure 2b) rather than the expected flat trend. As more screenshots replace text documents (increasing $p$), performance initially drops—from 0.22 at $p = 0$ (all text) to 0.02 at $p = 0.99$ (99% screenshots). However, at $p = 1$ (all screenshots), performance improves again to 0.36, forming a clear U-shape as a function of $p$. Interestingly, CLIP performs better on text-to-image retrieval ($p = 1$) than on text-to-text retrieval ($p = 0$), likely due to its training objective: cross-modal contrastive loss, without explicit optimization for unimodal retrieval.

**The U-shape arises from the modality gap.** We attribute the U-shaped performance to modality gap. First, the modality gap induces intra-modal similarity bias: although CLIP aligns text and image embeddings in a shared space, text and image clusters remain separate (Figure 1c), resulting in systematically higher intra-modal similarity scores (Figure 1d). Second, this bias causes ranking distortion. As screenshots replace more text entries, relevant screenshots are penalized due to lower cross-modal similarity, while irrelevant text documents may rank higher solely because of intra-modal alignment. At $p = 0.99$, the few remaining text documents dominate rankings regardless of relevance. At $p = 1$, all documents are images and modality bias disappears, leading to improved performance—thus forming the U-shaped curve.

**Push-down simulation confirms the hypothesis.** To verify this explanation, we simulate a modality-induced ranking bias by assigning a fixed similarity score of zero to all screenshots, effectively pushing them to the bottom of the ranked list. The resulting performance curve (Figure 2b) closely matches the actual CLIP curve, validating our hypothesis that the U-shape arises from modality gap–induced ranking distortion.

## 3.3 GR-CLIP WITH IMPROVED RESULTS

Given that the modality gap causes performance drops, we mitigate this gap to improve performance.

**Closing the modality gap via mean-shift calibration.** Following and adapting prior work characterizing the modality gap as a mean shift in the embedding space (Zhang et al., 2024), we propose GR-CLIP, a lightweight post-hoc calibration method. We compute the mean embeddings for text and image modalities and subtract them from their respective representations to center both modalities in the shared space. This reduces separation between modalities (Figure 2d; see §2 for derivation).

**Flattened curves and improved performance after removing the modality gap.** After applying GR-CLIP, retrieval performance improves significantly, and the U-shaped curve flattens across different $p$ values (Figure 2e). GR-CLIP also outperforms VLM2Vec (Jiang et al., 2025b), a recent generative embedding method that achieves similarly flat performance but requires $75\times$ more computational resources. These results demonstrate that reducing the modality gap is both efficient and effective for improving CLIP-based model in mixed modality retrieval settings.

## 3.4 GENERALIZATION ACROSS MODELS, DATASETS, AND MODALITIES

To assess the generality of our findings, we evaluate GR-CLIP across different models, datasets, and modalities. **1) Across models:** As shown in Figure 2f (top row), the U-shaped curve is observed across three CLIP variants: OpenAI CLIP (Radford et al., 2021), OpenCLIP (Cherti et al., 2023), and SigLIP (Zhai et al., 2023). GR-CLIP consistently flattens the curve and improves performance; **2) Across datasets:** As shown in Figure 2f (second row), our findings extend beyond synthetic screenshot settings (NFCorpus (Boteva et al., 2016) and SciFact (Wadden et al., 2020)) to real-world datasets (Google WIT (Srinivasan et al., 2021) and MSCOCO (Lin et al., 2014)); **3) Across modalities.** We also test generalization to text-to-video and text-to-audio retrieval (Appendix A).

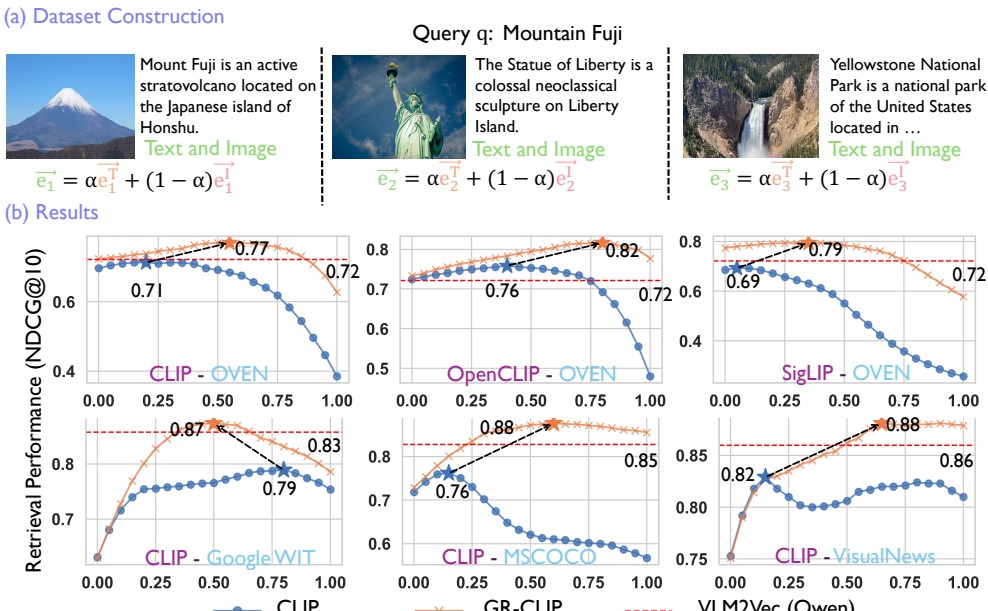

Figure 3: **Retrieval with multimodal documents.** **(a) Dataset Construction:** Each document contains both image and text, and embeddings are obtained by fusing modality-specific features. We vary the fusion coefficient $\alpha$ to evaluate the model's ability to integrate multimodal information. **(b) Results:** GR-CLIP consistently outperforms CLIP across three model variants and four datasets, demonstrating that the modality gap hinders effective multimodal fusion—and that removing it significantly enhances retrieval performance.

## 4 RETRIEVAL WITH MULTIMODAL DOCUMENTS

We now consider a complementary ablation to §3, where the retrieval corpus is homogeneous, but each document is multimodal—containing both image and text modalities (Figure 3a). This setup evaluates the model's ability to fuse multimodal information, where image and text together should provide richer semantic cues than either modality alone.

### 4.1 DATASET CONSTRUCTION

We use four real-world multimodal datasets in which each document contains both image and text components. **OVEN** (Hu et al., 2023) is an existing retrieval benchmark that follow a query-to-multimodal-document format, where each query itself is a text–image pair. For **MSCOCO** (Lin et al., 2014) and **VisualNews** (Liu et al., 2021), each image is paired with one or more short captions; we randomly sample one caption as the query and generate a long caption using GPT by conditioning on short captions along with the image to form the document. In **Google WIT** (Srinivasan et al., 2021), each image is accompanied by a title, a short caption, and a long caption. We use the concatenation of the title and short caption as the query, and the image combined with the long caption as the document. These datasets span diverse domains with naturally paired image-text data. Each document provides complementary visual and textual signals, making them well-suited for evaluating modality fusion. Additional details on dataset construction are provided in Appendix F.

### 4.2 RESULTS

To analyze how modality fusion is affected by the modality gap, we vary the fusion weight $\alpha \in [0, 1]$, which controls the contribution of each modality for the fused embedding: $e_i = \alpha \cdot e_i^T + (1 - \alpha) \cdot e_i^I$.

**Modality gap hinders effective fusion.** As shown in Figure 3b (blue curves), with the original CLIP embeddings, performance typically peaks at one of the unimodal endpoints ($\alpha = 0$ or $\alpha = 1$), and fusion with intermediate $\alpha$ values fails to outperform these unimodal baselines. This suggests that the modality gap prevents effective integration across modalities: linear interpolation often pushes the fused features into a suboptimal region in embedding space, degrading semantic quality and resulting in worse performance than using image or text alone.

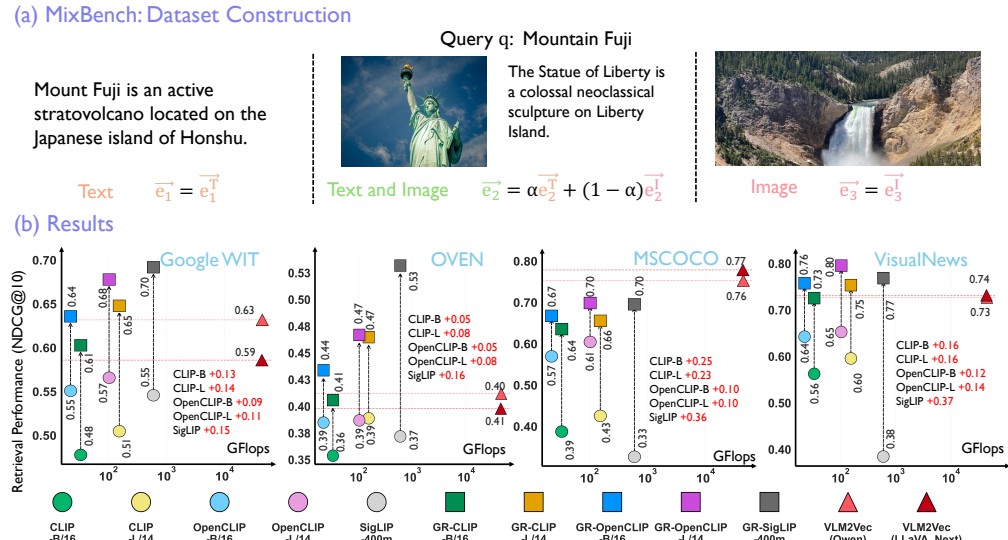

Figure 4: **Mixed modality search. (a) Dataset Construction:** We introduce **MixBench**, a benchmark where the corpus is heterogeneous and includes multimodal documents, reflecting the most realistic setting for search engines. **(b) Results:** Across four MixBench subsets and five CLIP variants, GR-CLIP delivers substantial improvements over the original CLIP models by eliminating the modality gap, achieving state-of-the-art performance with significantly lower computational cost.

**Fusion improves significantly after closing the modality gap.** Once the modality gap is removed (via mean-shift calibration as described in §3), fusion becomes substantially more effective. As shown in Figure 3b (orange curves), performance peaks at intermediate $\alpha$ values—surpassing both unimodal baselines. This demonstrates that the gap-removed model, GR-CLIP, successfully integrates complementary information from image and text, yielding stronger overall representations.

**Generalization across models and datasets.** These findings hold consistently across multiple CLIP variants, including OpenAI CLIP (Radford et al., 2021), OpenCLIP (Cherti et al., 2023), and SigLIP (Zhai et al., 2023), and across various datasets such as OVEN (Hu et al., 2023), Visual-News (Liu et al., 2021), Google WIT (Srinivasan et al., 2021), and MSCOCO (Lin et al., 2014). In all cases, removing the modality gap improves fusion quality, thereby enhancing retrieval performance.

## 5 MIXED MODALITY SEARCH

We now unify the findings from §3 and §4 and extend our analysis to the most realistic scenario: mixed modality search, where documents in the corpus may be purely text, purely image, or a combination of both (Figure 4a). This setting mirrors real-world search engine challenges, where retrieval systems must operate over heterogeneous and variably multimodal content.

### 5.1 MIXBENCH: DATASET CONSTRUCTION

To support research in this realistic setting, we introduce **MixBench**, a new benchmark specifically designed for mixed modality search. MixBench is constructed from four real-world multimodal datasets—**OVEN** (Hu et al., 2023), **MSCOCO** (Lin et al., 2014), **Google WIT** (Srinivasan et al., 2021), and **VisualNews** (Liu et al., 2021)—which span diverse domains and contain naturally aligned image-text content. The procedure for converting these datasets into a query-document retrieval format is detailed in §4. In MixBench, documents may consist of image-only, text-only, or image-text pairs (Figure 4a), where we sample in a 1:1:1 ratio to ensure a balanced distribution.

### 5.2 RESULTS

Figure 4b presents results on the four MixBench subsets using both the original CLIP variants and their gap-removed counterparts (GR-CLIP).

**GR-CLIP shows substantial improvement over original CLIP after closing the modality gap.** Consistent with our earlier findings, closing the modality gap via mean-shift calibration leads to

significant performance improvements on MixBench across all tested models, including OpenAI CLIP (Radford et al., 2021), OpenCLIP (Cherti et al., 2023), and SigLIP (Zhai et al., 2023). These improvements generalize across the four datasets—OVEN (Hu et al., 2023), VisualNews (Liu et al., 2021), Google WIT (Srinivasan et al., 2021), and MSCOCO (Lin et al., 2014). On average, GR-CLIP achieves up to a 26 percentage point gain in NDCG@10, with negligible additional compute cost. These gains are driven by improved cross-modal alignment and multimodal fusion, as demonstrated in §3 and §4, which are critical for performance in mixed modality retrieval.

**GR-CLIP achieves state-of-the-art performance with significantly lower compute.** Notably, GR-CLIP outperforms the strong VLM2Vec baseline despite using $75\times$ fewer computational resources. The only exception is MSCOCO, which VLM2Vec has been trained on, as reported in the paper. These results underscore the importance of constructing a truly shared embedding space for mixed modality search—a capability that is essential for effective retrieval systems yet often overlooked.

## 6 RELATED WORK

**Unimodal, cross-modal and multimodal retrieval.** Unimodal retrieval (e.g., text-to-text, image-to-image), cross-modal retrieval (e.g., text-to-image, image-to-text) and multimodal retrieval have been extensively studied in prior work (Robertson et al., 2009; Karpukhin et al., 2020; Khattab & Zaharia, 2020; Lee et al., 2018; Radford et al., 2021; Zhang et al., 2025; Jiang et al., 2025a) and now power many search engines such as Google and Bing. The core challenge in these settings is to construct an effective representation space that enables accurate similarity comparison between queries and documents. In contrast, we focus on the more complex mixed modality retrieval setting, where both queries and documents may span multiple modalities (Voyage AI, 2024). This setting is significantly underexplored but highly practical. It presents a new challenge: designing a shared representation space where semantic similarities can be meaningfully measured across modality boundaries.

**Multimodal representation learning.** Multimodal representation learning aims to unify inputs from different modalities into a coherent embedding space, with early work exploring early fusion and late fusion techniques (Ngiam et al., 2011; Li et al., 2020; Chen et al., 2020). Recently, multimodal contrastive learning has emerged as a powerful framework, aligning paired image-text representations through contrastive objectives (Girdhar et al., 2023; Radford et al., 2021; Zhai et al., 2023; Cherti et al., 2023). Models like CLIP (Radford et al., 2021), trained on millions of paired examples, have shown remarkable ability to learn semantically aligned embeddings across modalities. More recently, there is growing interest in adapting generative vision-language models (VLMs) for retrieval (Jiang et al., 2025b; Faysse et al., 2025), by repurposing them as embedding models (BehnamGhader et al., 2024; Muennighoff et al., 2024). These models are more flexible and capable of handling diverse multimodal inputs, but often require significantly more computation. In this work, we evaluate both paradigms—CLIP (Radford et al., 2021) and VLM2Vec (Jiang et al., 2025b)—under the mixed modality retrieval setting. Surprisingly, we find that a simple calibration method applied to CLIP can outperform VLM2Vec, despite using far less compute.

**Modality gap in multimodal contrastive learning.** Recent studies (Liang et al., 2022; Zhang et al., 2024; 2023) have revealed a persistent modality gap in contrastive multimodal embedding spaces: image and text embeddings tend to cluster separately, even though contrastive learning is designed to align them. This gap has been attributed to a combination of model initialization and contrastive optimization. Theoretically, the modality gap has been characterized as a constant vector orthogonal to both the image and text subspaces (Zhang et al., 2024; 2023). Meanwhile, several training-based methods (Eslami & de Melo, 2025; An et al., 2025; Liu et al., 2023b) have also been proposed to reduce this gap by modifying the CLIP training objective. Building on these insight, we adopt a simple but effective mean-reduction calibration, which removes the modality-specific means from embeddings before computing similarity. This lightweight, post-hoc procedure removes the modality gap and leads to substantial gains in the mixed modality search setting.

## 7 CONCLUSION

This work addresses the realistic yet underexplored problem of mixed modality search, where queries must retrieve semantically relevant content from a heterogeneous corpus containing multimodal documents. We analyze the behavior of CLIP-based models in this setting and identify a key limitation: a modality gap in the embedding space hinders both cross-modal alignment and multimodal fusion. To address this, we introduce **GR-CLIP**, a simple yet effective method that removes the modality gap and substantially improves retrieval performance. Our findings highlight the importance of truly unified multimodal representations for reliable and efficient mixed modality search.

ETHICS STATEMENT

All authors have read and agreed to comply with the ICLR Code of Ethics (`https://iclr.cc/public/CodeOfEthics`). The datasets used in this work are either publicly available or adapted from public sources solely for research purposes, without including any personally identifiable or sensitive information. All datasets comply with their original licenses.

REPRODUCIBILITY STATEMENT

We are committed to ensuring the reproducibility of our results. The problem formulation, methodology, and evaluation setup are fully documented in the main paper and appendix. To further support replication and extension of our work, we release both the implementation code and datasets to enable other researchers to replicate and extend our findings.

**Code Availability** All code is available at an anonymous GitHub repository, which reproduces all experiments in the paper: `https://anonymous.4open.science/r/GR-CLIP-mixed-modality-search/`.

**Data Availability** All datasets used in this study are hosted anonymously on Hugging Face to facilitate future research in this emerging area: `https://huggingface.co/datasets/iclr2026-anonymous/MixBench2025`.

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

## LIMITATIONS

While our work demonstrates that removing the modality gap enables GR-CLIP to achieve substantial performance gains in the mixed modality search setting across diverse datasets, model variants, and modalities, several limitations remain, highlighting valuable directions for future research. First, although we consider a realistic scenario in which documents include both image and text modalities, each document is restricted to a single image and a single text segment. Extending the evaluation to more complex, interleaved multi-image and multi-text documents—such as web pages or scientific articles—could provide a more rigorous and comprehensive assessment. Second, although GR-CLIP outperforms the generative embedding model VLM2Vec while requiring significantly less computation, it builds on CLIP, which does not model fine-grained modality interaction, and may miss opportunities for deeper cross-modal integration that generative embedding models can capture. Given this, investigating the causes of the modality gap in generative embedding models such as VLM2Vec and developing methods to reduce it presents an important and underexplored research direction toward more powerful and unified multimodal representations.

Nonetheless, our work makes several significant contributions: (1) we are the first to formally define and investigate the mixed modality retrieval setting, where documents of different modalities are retrieved for the same query; (2) we identify the modality gap as a key challenge degrading performance; (3) we propose a simple, theoretically grounded post-hoc calibration that effectively mitigates this gap, consistently boosting performance across CLIP variants and datasets; (4) we introduce MixBench, a benchmark for evaluating retrieval in mixed-modality contexts. Together, these contributions lay a foundation for future advances in this important, emerging area.

## USE OF LARGE LANGUAGE MODELS

We used large language models (LLMs) only for language editing and grammar improvement of the manuscript. No LLMs were used for research ideation, experimental design, analysis, or writing of technical content.

## OVERVIEW

We provide an overview of the Appendix below:

- §A presents additional generalization results across modalities and evaluation metrics.
- §B demonstrates the robustness of our method under different settings, including supervised fine-tuning, mean computation, and benchmarks.
- §C details the methods and includes pseudo-code for reproducibility.
- §D describes the details of the models used.
- §E explains the evaluation metrics, including NDCG.
- §F outlines the datasets used and the associated preprocessing steps.
- §G includes case studies comparing CLIP and GR-CLIP on MixBench.

# A    GENERALIZATION ACROSS MODALITIES AND METRICS

In the main paper, we show that closing the modality gap significantly improves mixed modality search performance for image-text data, using NDCG@10 as the evaluation metric. Here, we provide additional results to demonstrate: (1) the generalization of our method to modalities beyond image and text, and (2) the robustness of our conclusions under alternative evaluation metrics.

## A.1    GENERALIZATION ACROSS MODALITIES

Figure 2e in the main paper presents results for the image-text modality. In Figure 5, we extend this analysis to additional modality pairs. Specifically, we report retrieval performance (NDCG@10) for video-text (ViCLIP (Wang et al., 2022) on the MSVD dataset), audio-text (CLAP (Wu et al., 2023) on the Clotho (Drossos et al., 2020) dataset), and an additional image-text setting (OpenAI CLIP (Radford et al., 2021) on the Nights (Fu et al., 2023) dataset). Across all cases, we observe a consistent U-shaped curve in the original CLIP-based results, which becomes significantly flatter after applying GR-CLIP to remove the modality gap. This trend closely mirrors the behavior observed in the image-text and screenshot experiments in Figure 2e, providing strong evidence of the modality gap's impact and the broad applicability of our method across diverse modalities.

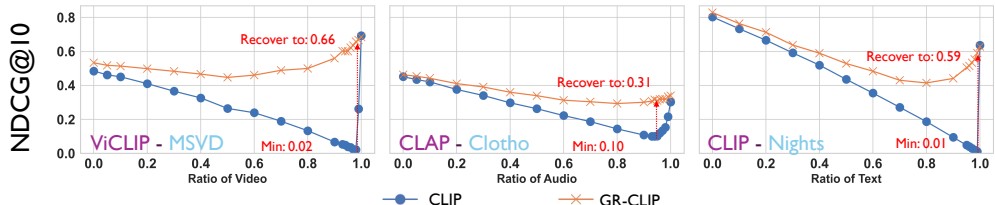

Figure 5: **Generalization across modalities.** GR-CLIP consistently mitigates the U-shaped curve caused by the modality gap and significantly improves performance, demonstrating strong generalizability across diverse modality pairs.

## A.2    GENERALIZATION ACROSS METRICS

In the main paper, we adopt NDCG@10 as the primary evaluation metric. To further assess the robustness of GR-CLIP, we extend our analysis to additional metrics, including NDCG@100 and Recall@1. Table 1 reports results on MixBench across all three metrics, demonstrating that the improvements observed with GR-CLIP are consistent regardless of the evaluation criterion. Figure 6 and Figure 7 further extend the analysis in §3 and §4 using NDCG@100 and Recall@1, respectively, and similarly confirm the consistency of our findings.

Additionally, we have also conducted a human study to evaluate different methods on MixBench. Specifically, we asked five independent users to judge which ranking they preferred; the GR-CLIP ranking was preferred in all cases (100%). This further confirms that GR-CLIP not only improves quantitative retrieval metrics but also better aligns with human perceptual judgments, reinforcing its practical utility in real-world mixed-modality search scenarios.

| Method | Google WIT | MSCOCO | OVEN | VisualNews |
|---|---|---|---|---|
| CLIP-B/16 | 0.478/0.505/0.443 | 0.388/0.426/0.292 | 0.354/0.398/0.209 | 0.563/0.604/0.498 |
| CLIP-L/14 | 0.505/0.516/0.454 | 0.426/0.490/0.329 | 0.389/0.431/0.253 | 0.596/0.656/0.525 |
| OpenCLIP-B/16 | 0.551/0.563/0.519 | 0.570/0.615/0.489 | 0.385/0.426/0.229 | 0.643/0.693/0.543 |
| OpenCLIP-L/14 | 0.566/0.585/0.536 | 0.605/0.662/0.540 | 0.387/0.445/0.265 | 0.653/0.733/0.567 |
| SigLIP-400m | 0.546/0.566/0.523 | 0.327/0.374/0.260 | 0.372/0.428/0.271 | 0.385/0.475/0.366 |
| VLM2Vec(LLaVANext) | 0.586/0.616/0.481 | **0.769/0.798/0.645** | 0.398/0.443/0.254 | 0.744/0.794/0.662 |
| VLM2Vec(Qwen) | 0.632/0.660/0.519 | 0.753/0.778/0.633 | 0.412/0.467/0.244 | 0.734/0.784/0.653 |
| GR-CLIP-B/16 | 0.603/0.642/0.524 | 0.636/0.690/0.523 | 0.406/0.459/0.240 | 0.726/0.768/0.645 |
| GR-CLIP-L/14 | 0.648/0.678/0.555 | 0.656/0.708/0.547 | 0.465/0.523/0.296 | 0.754/0.770/0.661 |
| GR-OpenCLIP-B/16 | 0.636/0.666/0.572 | 0.668/0.751/0.589 | 0.434/0.490/0.253 | 0.758/0.783/0.664 |
| GR-OpenCLIP-L/14 | 0.678/0.704/0.604 | 0.699/0.784/0.629 | 0.467/0.525/0.282 | **0.796/0.814/0.715** |
| GR-SigLIP-400m | **0.692/0.722/0.608** | 0.696/0.732/0.548 | **0.532/0.581/0.328** | 0.769/0.793/0.671 |

Table 1: **Detailed results across all metrics on MixBench.** Each cell reports NDCG@10, NDCG@100, and Recall@1. Best results are highlighted in bold. The consistent performance across metrics demonstrates the robustness of our approach to different evaluation criteria. GR-CLIP underperforms VLM2Vec on MSCOCO because VLM2Vec was trained on MSCOCO.

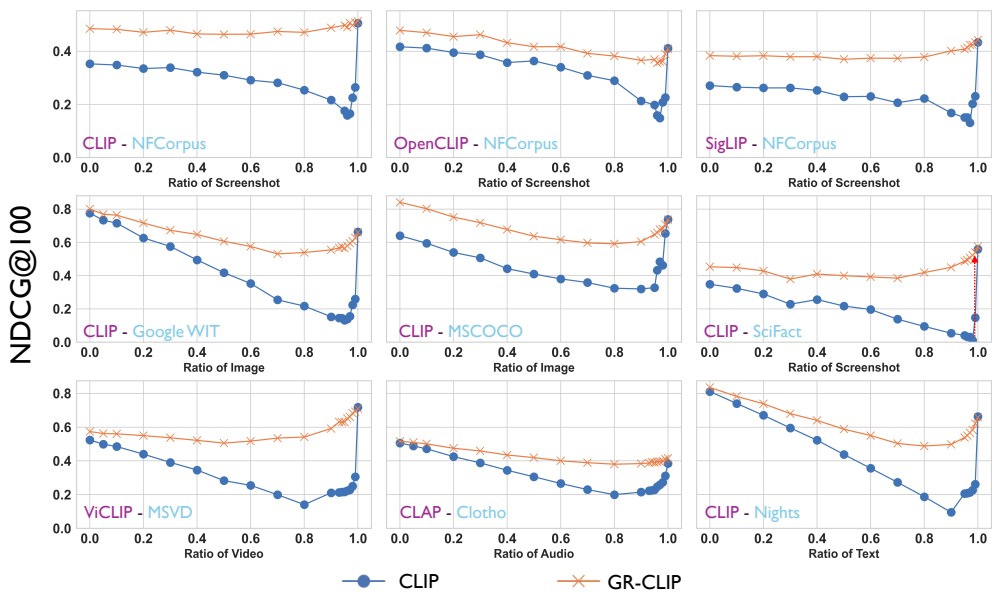

Figure 6: **Reproduction of Figure 2 in the main paper using NDCG@100 as the evaluation metric.**

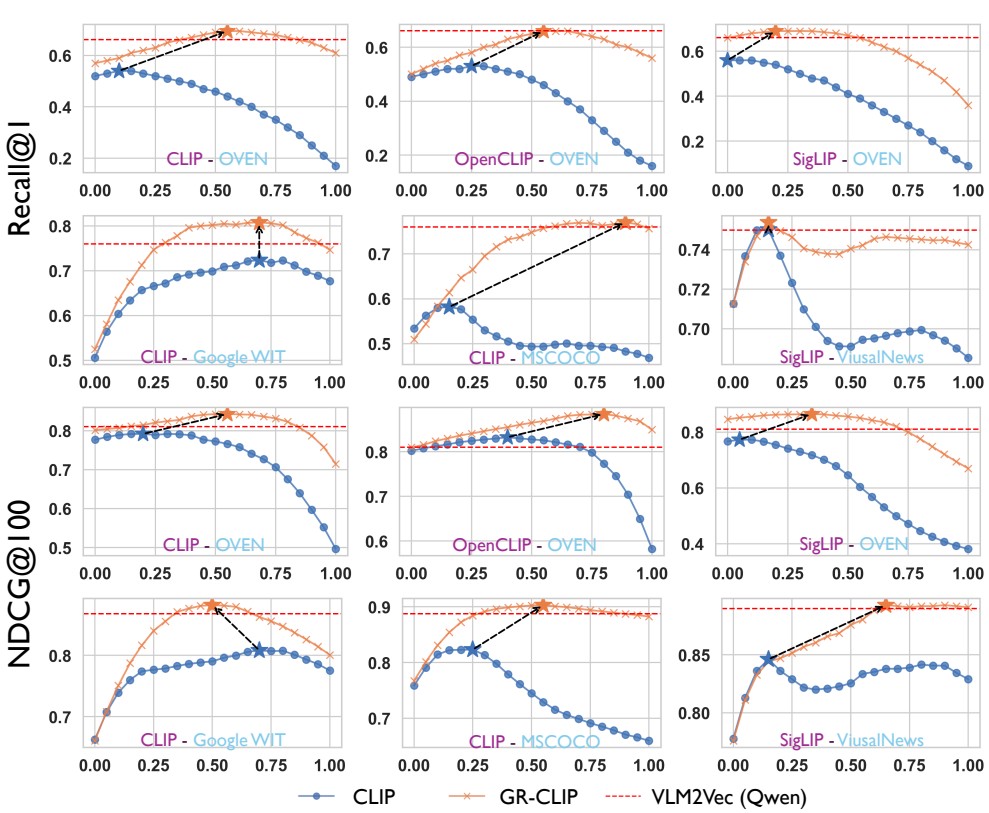

Figure 7: **Reproduction of Figure 3 in the main paper using NDCG@100 and Recall@1 as evaluation metrics.**

# B    METHOD ROBUSTNESS

## B.1    METHOD ROBUSTNESS TO SUPERVISED FINE-TUNING

To assess the applicability of our method after supervised fine-tuning, we fine-tuned the OpenAI CLIP model on the MSCOCO training set (denoted as "-SFT") using a batch size of 64, a learning rate of 5e-5, and the default temperature. We then applied our method to these fine-tuned models. The NDCG@10 scores on MixBench are shown in Table 2.

| Model | Google WIT | MSCOCO | OVEN | VisualNews |
|---|---|---|---|---|
| CLIP-B-SFT | 0.489 | 0.513 | 0.358 | 0.588 |
| GR-CLIP-B-SFT | 0.617 | 0.727 | 0.412 | 0.745 |
| CLIP-L-SFT | 0.514 | 0.538 | 0.396 | 0.612 |
| GR-CLIP-L-SFT | 0.656 | 0.792 | 0.474 | 0.772 |

Table 2: **GR-CLIP remains effective with supervised fine-tuned models.**

As Table 2 shows, our method consistently improves retrieval performance across all domains, even after supervised fine-tuning, suggesting that the modality gap persists and that our approach remains effective in practical deployment scenarios.

## B.2    METHOD ROBUSTNESS TO MEAN-EMBEDDING COMPUTATION

To examine our method's robustness to mean-embedding computation, we recomputed the mean embeddings without using any MSCOCO data and then applied GR-CLIP to the MSCOCO dataset. As shown in Table 3, consistent performance improvements were observed across models. Each cell reports NDCG@10 for the base CLIP model, GR-CLIP with MSCOCO access, and GR-CLIP using means computed without MSCOCO. Importantly, the gains persist even when MSCOCO itself is excluded from the mean computation, demonstrating that our approach requires access only to data with the same modality and a similar style, not necessarily the same dataset.

| Model | CLIP-B | CLIP-L | OpenCLIP-B | OpenCLIP-L | SigLIP |
|---|---|---|---|---|---|
| CLIP | 0.388 | 0.426 | 0.570 | 0.605 | 0.327 |
| GR-CLIP | 0.636 | 0.656 | 0.668 | 0.699 | 0.696 |
| GR-CLIP (w/o access to MSCOCO) | 0.624 | 0.648 | 0.659 | 0.687 | 0.683 |

Table 3: **NDCG@10 on MSCOCO using mean embeddings estimated with and without MSCOCO.** GR-CLIP improves over CLIP even without direct access to MSCOCO during mean computation.

We further assessed cross-dataset transfer by computing the query, text, and image embedding means using only 1,000 items from MSCOCO and applying them to VisualNews. As shown in Table 4, we still observed meaningful gains, indicating that our mean-estimation procedure is lightweight, data-efficient, and generalizable across datasets.

| Model | CLIP-B | CLIP-L | OpenCLIP-B | OpenCLIP-L | SigLIP |
|---|---|---|---|---|---|
| CLIP | 0.563 | 0.596 | 0.643 | 0.653 | 0.385 |
| GR-CLIP | 0.726 | 0.754 | 0.758 | 0.796 | 0.769 |
| GR-CLIP (using 1k items in MSCOCO) | 0.678 | 0.694 | 0.712 | 0.742 | 0.732 |

Table 4: **NDCG@10 on VisualNews using mean embeddings estimated from MSCOCO.** GR-CLIP maintains strong performance even when the means are estimated from a small subset (1,000 samples).

## B.3 METHOD ROBUSTNESS ON BENCHMARKS

To further demonstrate the robustness of GR-CLIP, we evaluated it on the MMEB benchmark, which contains subsets involving both text–only and image–text retrieval tasks. Unlike our main setting, which focuses on mixed-modality retrieval, MMEB assumes homogeneous corpora in which all candidates share the same modality. Nevertheless, as shown in Table 5, GR-CLIP consistently improves upon CLIP-family baselines across a wide range of subsets. The improvements are particularly strong on datasets containing image–text pairs, such as VisualNews, MSCOCO, and WebQA.

| Subset | CLIP-B | CLIP-L | OpenCLIP-B | OpenCLIP-L | SigLIP | VLM2Vec |
|---|---|---|---|---|---|---|
| VisualNews_t2i | 74.1 / 78.1 | 76.4 / 79.6 | 75.4 / 79.3 | 79.4 / **82.3** | 52.4 / 57.7 | 74.4 |
| VisualNews_i2t | 73.2 / 77.0 | 81.3 / 85.3 | 79.5 / 84.2 | 82.4 / **86.4** | 51.3 / 56.4 | 80.1 |
| MSCOCO_t2i | 55.3 / 58.2 | 58.0 / 63.3 | 62.4 / 66.5 | 66.5 / 70.9 | 58.3 / 64.4 | **74.5** |
| MSCOCO_i2t | 52.8 / 57.4 | 56.1 / 61.2 | 60.3 / 64.2 | 63.2 / 68.4 | 54.3 / 65.5 | **72.8** |
| Nights | 56.3 / 60.1 | 67.1 / 69.0 | 61.7 / 67.1 | 70.6 / **73.2** | 63.4 / 68.9 | 65.5 |
| WebQA | 61.0 / 78.3 | 62.4 / 80.4 | 64.8 / 82.5 | 65.2 / 84.3 | 59.7 / 71.3 | **86.6** |
| OVEN | 42.3 / 66.5 | 48.5 / 67.9 | 48.7 / 69.8 | 50.3 / **71.2** | 55.3 / 70.7 | 56.3 |
| FashionIQ | 9.7 / 11.5 | 13.9 / 14.9 | 12.2 / 16.9 | 15.4 / 16.7 | 20.3 / **20.8** | 16.1 |
| EDIS | 75.3 / 90.1 | 78.9 / 90.7 | 80.1 / **93.5** | 82.3 / 92.5 | 23.7 / 58.7 | 87.9 |
| WikiSS-NQ | 50.6 / 54.2 | 46.4 / 49.4 | 54.3 / 57.8 | 44.0 / 46.2 | 56.3 / **59.4** | 58.8 |
| Average (Multimodal Doc Subsets) | 59.5 / 78.3 | 63.3 / 79.7 | 64.5 / 81.9 | 65.9 / **82.7** | 46.2 / 66.9 | 76.9 |
| Average (All) | 55.1 / 63.4 | 59.5 / 66.4 | 59.4 / 67.7 | 61.9 / **69.2** | 49.5 / 60.0 | 67.3 |

Table 5: **Recall@1 results on MMEB datasets.** Each entry is shown as base model / GR-model. GR-CLIP consistently improves CLIP-family models, with the strongest gains on image-text retrieval datasets. Note that VLM2Vec is trained on MSCOCO and WebQA, whereas the CLIP models are not; this in-domain training likely accounts for its superior performance on these benchmarks.

GR-CLIP is evaluated in a zero-shot setting, whereas VLM2Vec is fine-tuned. Subsets such as VisDial and CIRR require dialog understanding or instruction following—tasks that go beyond CLIP's training paradigm. Hence, modality-gap removal alone cannot bridge those task-specific gaps.

### B.4 METHOD ROBUSTNESS TO DIFFERENT DISTRIBUTION OF DOCUMENT MODALITY IN MIXBENCH

Our choice of an equal proportion of image-only, text-only, and multimodal documents is intentional: we aim to simulate a setting where modalities have equal weight, and the corpus does not itself introduce a modality-frequency gap. This design removes distribution-level confounders and allows us to isolate the effect of modality misalignment in the embeddings rather than the effect of corpus imbalance.

To further demonstrate the robustness of GR-CLIP under different modality distributions—including a more realistic skewed setting (text:image: multimodal = 7:2:1), we rebuilt Mixbench with that distribution and evaluated GR-CLIP on it. Table 6 shows that GR-CLIP continues to deliver consistent gains and outperforms VLM embedding models. This further confirms that our method is robust to corpus-level modality ratios.

### B.5 COMPARISON WITH OTHER CALIBRATION METHODS

Although our goal is not to develop an optimal modality-gap calibration method, we include comparisons with training-based approaches for completeness. In our work, we treat AlignCLIP (Eslami & de Melo, 2025), I0T (An et al., 2025), UniVL-DR (Liu et al., 2023b) as additional baselines and evaluate them on MixBench. As shown in Table 6, GR-CLIP-B outperforms UniVL-DR-CLIP-B and AlignCLIP-CLIP-B and I0T-CLIP-B-MSCOCO, indicating that GR-CLIP serves as a more efficient approach for mitigating the modality gap. Moreover, it is worth noting that UniVL-DR, I0T, and AlignCLIP are training-based approaches, whereas GR-CLIP is entirely training-free, which makes our method significantly more lightweight and easier to deploy in practice.

| Models | Google-WIT | MSCOCO | OVEN | VisualNews | GFLOPS* |
|---|---|---|---|---|---|
| CLIP-B | 0.478 / 0.597 | 0.388 / 0.508 | 0.354 / 0.265 | 0.563 / 0.653 | – |
| CLIP-L | 0.505 / 0.616 | 0.426 / 0.502 | 0.389 / 0.307 | 0.596 / 0.702 | – |
| OpenCLIP-B | 0.551 / 0.664 | 0.570 / 0.673 | 0.385 / 0.289 | 0.643 / 0.722 | – |
| OpenCLIP-L | 0.566 / 0.687 | 0.605 / 0.689 | 0.387 / 0.299 | 0.653 / 0.728 | – |
| SigLIP | 0.546 / 0.692 | 0.327 / 0.423 | 0.372 / 0.306 | 0.385 / 0.501 | – |
| AlignCLIP-CLIP-B-MSCOCO | 0.532 / 0.656 | 0.603 / 0.671 | 0.353 / 0.284 | 0.589 / 0.672 | – |
| AlignCLIP-CLIP-B-Flickr | 0.543 / 0.671 | 0.618 / 0.693 | 0.348 / 0.273 | 0.603 / 0.682 | – |
| I0T-CLIP-B-MSCOCO | 0.526 / 0.624 | 0.633 / 0.709 | 0.363 / 0.282 | 0.587 / 0.672 | – |
| UniVL-DR-CLIP-B | 0.495 / 0.623 | 0.403 / 0.488 | 0.363 / 0.342 | 0.586 / 0.703 | – |
| VLM2Vec (LLaVA-Next) | 0.586 / 0.709 | 0.769 / 0.841 | 0.398 / 0.359 | 0.744 / 0.823 | 75x |
| VLM2Vec (Qwen) | 0.632 / 0.742 | 0.753 / 0.833 | 0.412 / 0.387 | 0.734 / 0.815 | 75x |
| GR-CLIP-B | 0.603 / 0.712 | 0.636 / 0.721 | 0.406 / 0.332 | 0.726 / 0.799 | – |
| GR-CLIP-L | 0.648 / 0.765 | 0.656 / 0.748 | 0.465 / 0.425 | 0.754 / 0.824 | – |
| GR-OpenCLIP-B | 0.636 / 0.772 | 0.668 / 0.803 | 0.434 / 0.407 | 0.758 / 0.836 | – |
| GR-OpenCLIP-L | 0.678 / 0.792 | 0.699 / 0.826 | 0.467 / 0.434 | 0.796 / 0.862 | – |
| GR-SigLIP | 0.692 / 0.771 | 0.696 / 0.834 | 0.532 / 0.512 | 0.769 / 0.827 | – |

Table 6: NDCG@10 scores on MixBench under balanced (text:image: multimodal = 1:1:1) and skewed (text: image:multimodal = 7:2:1) modality distributions. GFLOPS* is measured as the ratio relative to GR-SigLIP

## C    DETAILS OF METHODS

As introduced in §2.3, GR-CLIP mitigates the modality gap by subtracting global mean vectors for each modality. Specifically, we compute three mean vectors: the query mean $\bar{e}_q$, the document text mean $\bar{e}^T$, and the document image mean $\bar{e}^I$:

$$\bar{e}_q = \mathbb{E}_{q\sim\mathcal{Q}}[f^T(q)], \quad \bar{e}^T = \mathbb{E}_{d^T\sim\mathcal{D}_{\text{text}}}[f^T(d^T)], \quad \bar{e}^I = \mathbb{E}_{d^I\sim\mathcal{D}_{\text{image}}}[f^I(d^I)]. \tag{1}$$

We distinguish the query mean $\bar{e}_q$ from the text document mean $\bar{e}^T$ to account for structural and semantic differences: queries are often short and interrogative, whereas documents are typically longer and descriptive. This distinction is crucial for reducing alignment bias and improving retrieval performance.

To ensure generalization across datasets and prevent test-set leakage, we compute unified mean vectors from the training sets of multiple datasets, rather than estimating separate means for each dataset using their respective test sets. These unified means are then applied consistently across all test sets.

**Query mean ($\bar{e}_q$):** We sample approximately 10000 text queries from the training splits of MSCOCO, Google WIT, NFCorpus, and VisualNews. These are encoded using $f^T$ and averaged to produce the global query mean $\bar{e}_q$.

**Document text mean ($\bar{e}^T$):** We sample approximately 10000 long-form text documents or descriptive captions from the training splits of MSCOCO, OVEN, Google WIT, and VisualNews. These are encoded using $f^T$ and averaged to obtain the document text mean $\bar{e}^T$.

**Document image mean ($\bar{e}^I$):** To compute $\bar{e}^I$, we sample 10000 images from the training splits of MSCOCO, OVEN, Google WIT, and VisualNews. These are encoded using $f^I$ and averaged to produce the document image mean.

**OVEN-Specific Query Mean ($\bar{e}_q^{\text{OVEN}}$):** Since queries in OVEN are particularly short, we construct a dataset-specific query mean by sampling 2000 queries from the OVEN training split.

**Other Modality Means:** For non-image-text datasets—such as MSVD (video-text), Clotho (audio-text), and screenshot-style documents (screenshot-text) in SciFact and NFCorpus—we compute modality-specific means using 2500 training examples per modality.

We summarize the full **GR-CLIP** algorithm as follows:

---

**Algorithm 1 GR-CLIP** Algorithm

---

**Require:**
1: Calibration sets: $\mathcal{Q}', \mathcal{D}'$
2: Query set $\mathcal{Q} = \{q_1, \ldots, q_n\}$ (text only)
3: Document set $\mathcal{D} = \{d_1, \ldots, d_m\}$ (text, image, or both for each)
4: Pretrained encoders $f^T, f^I$, interpolation factor $\alpha \in [0, 1]$
   *// Step 1: Pre-compute global means from $\mathcal{Q}', \mathcal{D}'$*
5: $\bar{e}_q \leftarrow \mathbb{E}_{q\sim\mathcal{Q}'}[f^T(q)]$
6: $\bar{e}^T \leftarrow \mathbb{E}_{d^T\sim\mathcal{D}'_{\text{text}}}[f^T(d^T)]$
7: $\bar{e}^I \leftarrow \mathbb{E}_{d^I\sim\mathcal{D}'_{\text{image}}}[f^I(d^I)]$
   *// Step 2: Encode query embeddings*
8: **for all** $q_i \in \mathcal{Q}$ **do**
9:     $e_{q_i} \leftarrow f^T(q_i) - \bar{e}_q$
10: **end for**

*// Step 3: Encode document embeddings*
11: **for all** $d_j \in \mathcal{D}$ **do**
12:     **if** $d_j$ is text **then**
13:         $e_{d_j} \leftarrow f^T(d_j) - \bar{e}^T$
14:     **else if** $d_j$ is image **then**
15:         $e_{d_j} \leftarrow f^I(d_j) - \bar{e}^I$
16:     **else if** $d_j = (d_j^T, d_j^I)$ **then**
17:         $e_{d_j} \leftarrow \alpha f^T(d_j^T) + (1-\alpha)f^I(d_j^I)$
18:             $-[\alpha\bar{e}^T + (1-\alpha)\bar{e}^I]$
19:     **end if**
20: **end for**
   *// Step 4: Retrieval*
21: $s(q_i, d_j) \leftarrow \frac{e_{q_i} \cdot e_{d_j}}{\|e_{q_i}\| \cdot \|e_{d_j}\|}$
22: Ranks $\leftarrow$ argsort$(s, \text{descending})$
23: **return** Ranks

---

## D  DETAILS OF MODELS

In this section, we provide the exact versions and checkpoint links for all models used in our experiments. For CLIP-based models, we include two variants of **OpenAI CLIP** (Radford et al., 2021), two variants of **OpenCLIP** (Cherti et al., 2023), and **SigLIP-400M** (Zhai et al., 2023).

For the VLM2Vec framework, we use two variants: one based on **LLaVA-Next** (Liu et al., 2024), which serves as the backbone for the results reported in the main paper (Jiang et al., 2025b); and another based on the latest officially released **Qwen-VL** (Bai et al., 2023), which achieves the best performance on the MMEB (Jiang et al., 2025b) benchmark according to its official repository.

Additionally, for non-image-text modalities, we use **ViCLIP**(Wang et al., 2022) for video-text retrieval and **CLAP**(Wu et al., 2023) for audio-text retrieval tasks.

All model checkpoint links are listed below:

- **OpenAI CLIP-B/16**: https://huggingface.co/openai/clip-vit-base-patch16
- **OpenAI CLIP-L/14**: https://huggingface.co/openai/clip-vit-large-patch14-336
- **OpenCLIP-B/16**: https://huggingface.co/laion/CLIP-ViT-B-16-laion2B-s34B-b88K
- **OpenCLIP-L/14**: https://huggingface.co/laion/CLIP-ViT-L-14-laion2B-s32B-b82K
- **SigLIP-400m**: https://huggingface.co/google/siglip-so400m-patch14-384
- **VLM2Vec (LLaVA-Next)**: https://huggingface.co/TIGER-Lab/VLM2Vec-LLaVa-Next
- **VLM2Vec (Qwen-VL)**: https://huggingface.co/TIGER-Lab/VLM2Vec-Qwen2VL-7B
- **ViCLIP-L/14**: https://huggingface.co/OpenGVLab/ViCLIP-L-14-hf
- **CLAP**: https://huggingface.co/laion/clap-htsat-fused

# E DETAILS OF EVALUATION METRICS

In the main paper, we use the widely adopted NDCG@10 as the evaluation metric. Here, we provide the detailed computation process for this metric.

Given a ranked list of retrieved items up to position $K$, NDCG@$K$ is computed as:

$$\text{NDCG@}K = \frac{1}{\text{IDCG@}K} \sum_{i=1}^{K} \frac{2^{\text{rel}_i} - 1}{\log_2(i+1)} \tag{2}$$

where $\text{rel}_i$ denotes the relevance score of the item at rank $i$, and IDCG@$K$ is the ideal DCG—that is, the maximum possible DCG for the top $K$ items—computed by sorting the items by relevance in descending order:

$$\text{IDCG@}K = \sum_{i=1}^{K} \frac{2^{\text{rel}_i^\star} - 1}{\log_2(i+1)} \tag{3}$$

where $\text{rel}_i^\star$ is the $i$-th highest relevance score in the ideal ranking.

NDCG@10 ranges from 0 to 1, with 1 indicating a perfect ranking.

# F  DETAILS OF DATASETS

In this section, we provide additional details on how each dataset is processed to support our retrieval experiments in §3, 4, and 5. For each dataset, we distinguish between the original data format (*Before*) and the modified version used in our framework (*After*). We also describe the key post-processing steps.

**NFCorpus (Boteva et al., 2016), SciFact (Wadden et al., 2020):**
*Before:* A short text query paired with a relevant long text document.
*After:* We retain the short text query and render the long text document into a screenshot using OpenCV. This allows retrieval of either the original text document or its rendered screenshot given the query.

**Google WIT (Srinivasan et al., 2021):**
*Before:* Each sample includes a page title, a long page description, a reference image, and a reference description for the image.
*After:* We concatenate the page title and image reference description to form the query. The page description is used as the long text document, and the associated image serves as the image document.

**OVEN (Hu et al., 2023):**
*Before:* Each query consists of an image-text pair, and the retrieval target is also an image-description pair.
*After:* Since either the image or text component can independently answer the query, we treat both the image and caption as valid standalone documents. The query remains unchanged.

**MSCOCO (Lin et al., 2014):**
*Before:* Each image is paired with five captions.
*After:* One caption is sampled as the query. The remaining captions are used to construct a long-form description via GPT-4o, with the content of the sampled caption preserved. This long description becomes the text document, and the associated image serves as the image document.

**VisualNews (Liu et al., 2021):**
*Before:* Each image is paired with a short news-style caption.
*After:* We use GPT-4o to jointly analyze the image and its associated article from the original VisualNews dataset. Based on both the visual content and article text, GPT-4o generates a detailed descriptive paragraph that expands upon the original caption, which we use as the text document. The image serves as the image document, and the original caption is retained as the query.

**Clotho (Drossos et al., 2020):**
*Before:* Each audio clip is paired with several semantically similar captions.
*After:* One caption is selected as the query, and another semantically similar caption (chosen by GPT-4o) is used as the text document. The audio clip itself is used as the audio document.

**MSVD (Chen & Dolan, 2011):**
*Before:* Each video is paired with several semantically similar captions.
*After:* One caption is used as the query, and another semantically similar caption (chosen by GPT-4o) serves as the text document. The video is treated as the video document.

**Nights (Fu et al., 2023):**
*Before:* Each image is paired with a visually similar image.
*After:* One image is used as the query. GPT-4o observes this image and generates a concise title, which we use as the text document. The paired image serves as the image document.

**VLM2Vec input format:** For **VLM2Vec** (Jiang et al., 2025b), prompts are required to serve as instructions for generating embeddings. Specifically, for each *Query*, we use the prompt "`Retrieve a relevant item that represents: {Query}\n`" in settings 1 and 3, which involve retrieval from a heterogeneous corpus composed of multiple modalities. In Setting 2, where retrieval is over a homogeneous corpus of fused image-text pairs, we use "`Retrieve an image-description pair that represents: {Query}\n`". Documents follow the format specified in the original datasets.

**CLIP input format:** For **CLIP**-based models (Radford et al., 2021; Zhai et al., 2023; Cherti et al., 2023; Wang et al., 2022; Wu et al., 2023) and **GR-CLIP**, we do not apply any instructions.

Queries and documents are directly passed to the respective CLIP text and image encoders without modification.

Table 7 summarizes the key characteristics of each dataset, including the retrieval setting, the modality composition of queries and corpora, and the total number of evaluation examples.

| Dataset | Queries | Documents | Setting No. | # of Queries | # of Documents |
|---|---|---|---|---|---|
| Google WIT (Srinivasan et al., 2021) | T | T / I / I + T | 1,2,3 | 1000 | 4423 |
| OVEN (Hu et al., 2023) | T + I | T / I / I + T | 1,2,3 | 1000 | 1000 |
| MSCOCO (Lin et al., 2014) | T | T / I / I + T | 1,2,3 | 984 | 984 |
| VisualNews (Liu et al., 2021) | T | T / I / I + T | 1,2,3 | 981 | 981 |
| SciFact (Wadden et al., 2020) | T | T / S | 1 | 300 | 5183 |
| NFCorpus (Boteva et al., 2016) | T | T / S | 1 | 323 | 3633 |
| MSVD (Chen & Dolan, 2011) | T | T / V | 1 | 670 | 670 |
| Clotho (Drossos et al., 2020) | T | T / A | 1 | 1046 | 1046 |
| Nights (Fu et al., 2023) | I | I / T | 1 | 1000 | 1000 |

Table 7: **Overview of datasets used in our experiments.** For each dataset, we indicate the retrieval setting, the modalities involved in queries and documents (T = text, I = image, S = screenshot, V = video, A = audio), and the number of query-document pairs used for evaluation.

# G CASE STUDIES

Below, we present case studies from each subset of MixBench, which also serve as visualizations of our dataset. For each example query, we display the Top-5 retrieved results from both the baseline OpenAI CLIP-L/14 and our proposed GR-CLIP-L/14 model. Each retrieved document is annotated with its modality (text, image, or multimodal), its cosine similarity to the query, and whether it is a **ground-truth** relevant item.

These example results illustrate both the diversity of the MixBench datasets and the effectiveness of GR-CLIP in mixed modality search. Unlike the original CLIP model, which tends to retrieve documents matching the query's modality, GR-CLIP successfully bridges the modality gap, retrieving results that more accurately reflect the semantic intent of the query—regardless of modality.

## G.1 GOOGLE WIT

*Query:* List of Jews in sports, Nate Ebner

---

**CLIP Top-5 Results**

*Rank No.1*, *Cosine Similarity* = 0.5430, *Modality* = text

This is a list of individuals currently serving in the United States House of Representatives.

--------------------------------------------------------------------------------

*Rank No.2*, *Cosine Similarity* = 0.5355, *Modality* = text

This is a list of notable Austrians.

--------------------------------------------------------------------------------

*Rank No.3*, *Cosine Similarity* = 0.5227, *Modality* = text

This is a list of vehicles manufactured by the Buick Motor Division of General Motors.

--------------------------------------------------------------------------------

*Rank No.4*, *Cosine Similarity* = 0.5181, *Modality* = text

This is a list of notable alumni and faculty of Golden Gate University.

--------------------------------------------------------------------------------

*Rank No.5*, *Cosine Similarity* = 0.5101, *Modality* = text

Puthenchira is a village in Thrissur district in the state of Kerala, India.

---

**GR-CLIP Top-5 Results**

*Rank No.1*, *Cosine Similarity* = 0.3403, *Modality* = Image (**Ground Truth**)

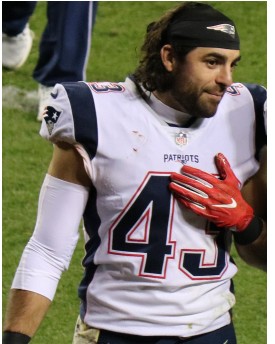

--------------------------------------------------------------------------------

*Rank No.2*, *Cosine Similarity* = 0.1798, *Modality* = text

This is a list of notable Austrians.

--------------------------------------------------------------------------------

*Rank No.3*, *Cosine Similarity* = 0.1774, *Modality* = text

The Lebanon national football team, controlled by the Lebanese Football Association, have represented Lebanon in association football since their inception in 1933. The squad is governed by the Asian Football Confederation continentally, and FIFA worldwide. While Lebanon have yet to qualify for the FIFA World Cup, they have participated twice in the Asian Cup: in 2000, when they hosted the event, and in 2019, the first time through regular qualification. Lebanon's main venue is the Camille Chamoun Sports City Stadium in Beirut; however they also play in other locations such as the Saida International Stadium in Sidon. In 1934, Lebanon played their first match against the Romanian side CA Timișoara, but it was not ratified by FIFA. Lebanon played their first FIFA-recognised game in 1940 against Mandatory Palestine. During their 2014 qualification campaign for the World Cup, Lebanon reached the final qualifying round for the first time thanks to a 2–1 victory against South Korea at home in 2011, but failed to qualify for the 2014 FIFA World Cup finishing bottom of their group. At the 2019 Asian Cup, Lebanon were close to qualifying to the knock-out stages for the first

time.

------------------------------------------------------------------------------------------------

*Rank No.4, Cosine Similarity* = 0.1723, *Modality* = text

This is a list of properties and historic districts in Winchester, Massachusetts, that are listed on the National Register of Historic Places. The locations of National Register properties and districts may be seen in an online map by clicking on "Map of all coordinates." This National Park Service list is complete through NPS recent listings posted July 17, 2020.

------------------------------------------------------------------------------------------------

*Rank No.5, Cosine Similarity* = 0.1708, *Modality* = multimodal

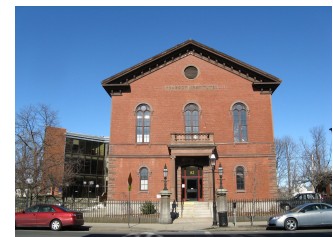

This list is of that portion of the National Register of Historic Places designated in Essex County, Massachusetts. The locations of these properties and districts for which the latitude and longitude coordinates are included below, may be seen in a map. There are more than 450 designated properties in the county, including 25 that are further designated as National Historic Landmarks. The municipalities of Andover, Gloucester, Ipswich, Lawrence, Lynn, Methuen, and Salem are to be found on a separate list of the more than 200 identified here, except two properties are split between Methuen and Lawrence, and one between Lynn and Nahant; these entries appear on more than one list. This National Park Service list is complete through NPS recent listings posted August 14, 2020.

## G.2 MSCOCO

*Query:* A woman in a room with a cat.

---

**CLIP Top-5 Results**

*Rank No.1*, *Cosine Similarity* = 0.5044, *Modality* = text

A kitchen featuring light wood cabinets and a black granite countertop. It includes a black stove with four burners, an over-the-range microwave, and a black refrigerator. The flooring is a warm wooden tone.

------------------------------------------------------------------------------------------------

*Rank No.2*, *Cosine Similarity* = 0.4605, *Modality* = text

A cat is perched on the closed lid of a toilet, appearing somewhat perturbed. The toilet is located in a bathroom with a light-colored wall. Next to the toilet, there is a basket or container. The cat's tail is visible, and it seems to be alert or possibly startled.

------------------------------------------------------------------------------------------------

*Rank No.3*, *Cosine Similarity* = 0.4445, *Modality* = text

A long hot dog is placed in a bun on a white paper plate, which sits on a wooden table. The hot dog extends beyond the ends of the bun.

------------------------------------------------------------------------------------------------

*Rank No.4*, *Cosine Similarity* = 0.4160, *Modality* = multimodal

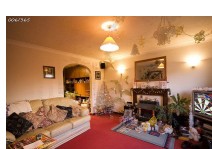
The warm and cozy living room is adorned with Christmas decorations, featuring a silver tinsel Christmas tree by the fireplace. The room is filled with a variety of gift-wrapped presents scattered around on the red carpet. On the mantelpiece, festive ornaments and stockings add to the holiday spirit. A comfortable beige sofa with cushions sits alongside a coffee table with magazines. The ceiling is decorated with shimmering golden stars, and a television displaying a dartboard game adds to the lived-in, festive atmosphere. The soft lighting from lamps enhances the room's inviting ambiance.

------------------------------------------------------------------------------------------------

*Rank No.5*, *Cosine Similarity* = 0.4126, *Modality* = text

A delicious Italian pizza is presented on a white plate, topped with slices of fresh tomatoes, green olives, and thinly sliced onions. The pizza is garnished with herbs and seasonings, adding a colorful and flavorful touch to the dish.

---

**GR-CLIP Top-5 Results**

*Rank No.1*, *Cosine Similarity* = 0.3012, *Modality* = multimodal (**Ground Truth**)

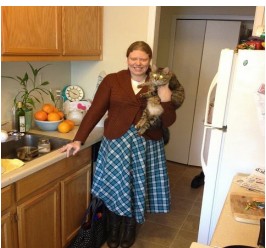
A woman is standing in a kitchen, smiling and holding a cat. She is wearing a brown sweater and a blue plaid skirt. The kitchen has wooden cabinets and a countertop with a potted plant and a bowl of oranges. There is a sink with dishes on one side and a white refrigerator on the other. A clock is visible on the wall, and there are various items on the counter and a small rug on the floor.

------------------------------------------------------------------------------------------------

*Rank No.2*, *Cosine Similarity* = 0.2924, *Modality* = multimodal

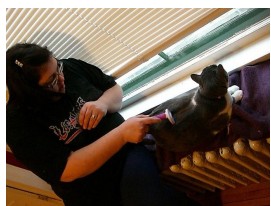
A person wearing glasses and a black shirt is sitting by a window with closed blinds, brushing a cat that is sitting on a purple blanket draped over a radiator. The cat is facing away, and the brush is Magenta with a grey bristle area. The floor is wooden, and the cat seems relaxed.

---------------------------------------------------------------------------------

*Rank No.3, Cosine Similarity* = 0.2780, *Modality* = text

A cat is perched on the closed lid of a toilet, appearing somewhat perturbed. The toilet is located in a bathroom with a light-colored wall. Next to the toilet, there is a basket or container. The cat's tail is visible, and it seems to be alert or possibly startled.

---------------------------------------------------------------------------------

*Rank No.4, Cosine Similarity* = 0.2745, *Modality* = multimodal

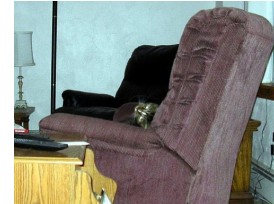

A gray armchair and a black armchair are positioned next to each other in a room. A small lamp is placed on a table next to the black chair. Partially visible from behind the armchair is a cat peeking out, adding a playful touch to the setting. In front of the chairs, there is a wooden table with a remote control on it.

---------------------------------------------------------------------------------

*Rank No.5, Cosine Similarity* = 0.2612, *Modality* = image

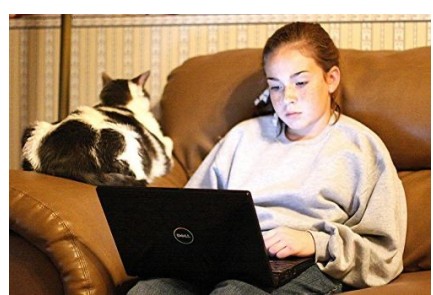

## G.3 OVEN

*Query:* 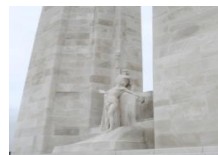 What is the name of this building?

---

**CLIP Top-5 Results**

*Rank No.1*, *Cosine Similarity* = 0.5340, *Modality* = multimodal

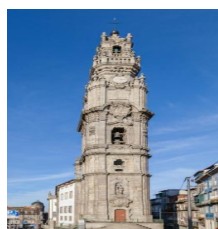 **Clérigos Church.** The Clérigos Church is a Baroque church in the city of Porto, in Portugal. Its 75-meter-tall bell tower, the Torre dos Clérigos, can be seen from various points of the city and is one of its most characteristic symbols. History: The church was built for the Brotherhood of the Clérigos (Clergy) by Nicolau Nasoni, an Italian architect and painter who left an extensive body of work in the north of Portugal during the 18th century. Construction of the church began in 1732 and was finished in 1750, while the bell tower and the monumental divided stairway...

*Rank No.2*, *Cosine Similarity* = 0.5321, *Modality* = image

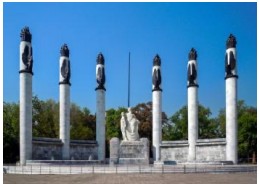

*Rank No.3*, *Cosine Similarity* = 0.5276, *Modality* = multimodal

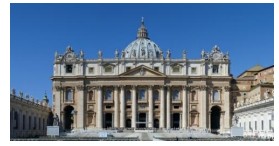 **St. Peter's Basilica.** The Papal Basilica of Saint Peter in the Vatican, or simply Saint Peter's Basilica, is a church built in the Renaissance style located in Vatican City. It was initially planned by Pope Nicholas V and then Pope Julius II to replace the aging Old St. Peter's Basilica, which was built in the fourth century by Roman emperor Constantine the Great. Construction of the present basilica began on 18 April 1506 and was completed on 18 November 1626. Designed principally by Donato Bramante, Michelangelo, Carlo Maderno, and Gian Lorenzo Bernini...

*Rank No.4*, *Cosine Similarity* = 0.5274, *Modality* = multimodal

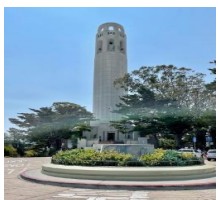 **Coit Tower.** Coit Tower is a 210-ft tower in the Telegraph Hill neighborhood of San Francisco, California, offering panoramic views over the city and the bay. Built between 1932 and 1933 using Lillie Hitchcock Coit's bequest to beautify the city, it was added to the National Register of Historic Places in 2008. The unpainted reinforced concrete tower, designed by Arthur Brown, Jr. and Henry Howard, features American fresco mural paintings by 25 different onsite artists...

*Rank No.5*, *Cosine Similarity* = 0.5252, *Modality* = multimodal

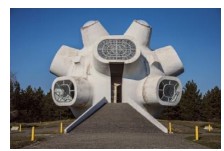 **Ilinden (Memorial).** Also known as Makedonium, Ilinden is a monument in Kruševo, North Macedonia. Officially opened on August 2, 1974, it commemorates the Second Session of the Anti-fascist Assembly and the 1903 Ilinden uprising. Designed by Jordan and Iskra Grabuloski, it honors fighters in the National Liberation Struggle from 1941–1944. Description. The monument covers 12 acres and features a rounded architectural style...

**GR-CLIP Top-5 Results**

*Rank No.1, Cosine Similarity* = 0.3153, *Modality* = text (**Ground Truth**)

Canadian National Vimy Memorial. The Canadian National Vimy Memorial is a war memorial site in France dedicated to the memory of Canadian Expeditionary Force members killed during the First World War. It also serves as the place of commemoration for Canadian soldiers of the First World War killed or presumed dead in France who have no known grave. The monument is the centrepiece of a 100 (ha) preserved battlefield park that encompasses a portion of the ground over which the Canadian Corps made their assault during the initial Battle of Vimy Ridge offensive of the Battle of Arras.

---

*Rank No.2, Cosine Similarity* = 0.2795, *Modality* = image

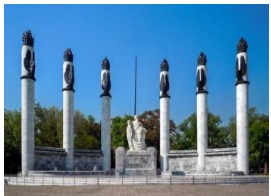

---

*Rank No.3, Cosine Similarity* = 0.2762, *Modality* = multimodal

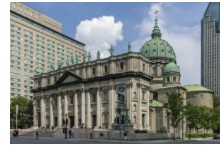 Mary, Queen of the World Cathedral. Mary, Queen of the World Cathedral or in full Mary, Queen of the World and St. James the Great Cathedral is a minor basilica in Montreal, Quebec, Canada, and the seat of the Roman Catholic archdiocese of Montreal. It is the third largest church in Quebec after Saint Joseph's Oratory (also in Montreal) and the Basilica of Sainte-Anne-de-Beaupré east of Quebec City. The building is 101 m (333 ft) in length, 46 m (150 ft) in width, and a maximum height of 77 m (252 ft) at the cupola, the diameter of which is 23 m (75 ft).

---

*Rank No.4, Cosine Similarity* = 0.2744, *Modality* = image

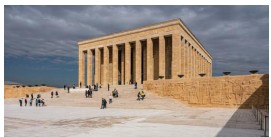

---

*Rank No.5, Cosine Similarity* = 0.2590, *Modality* = multimodal

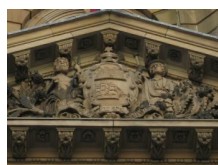 Sydney Town Hall. The Sydney Town Hall is a late 19th-century heritage-listed town hall building in the city of Sydney, the capital city of New South Wales, Australia, housing the chambers of the Lord Mayor of Sydney, council offices, and venues for meetings and functions. It is located at 483 George Street, in the Sydney central business district opposite the Queen Victoria Building and alongside St Andrew's Cathedral. Sited above the Town Hall station and between the city shopping and entertainment precincts, the steps of the Town Hall are a popular meeting place. It was designed by John H. Wilson, Edward Bell, Albert Bond.

### G.4 VISUALNEWS

*Query:* Former California officer Jay Cicinelli puts his head in his hands immediately after hearing the not guilty verdict in murder trial of a homeless man.

---

**CLIP Top-5 Results**

*Rank No.1*, *Cosine Similarity* = 0.4364, *Modality* = text

In this courtroom sketch, a solemn scene unfolds as the individual is depicted during the sentencing phase of a high-profile trial. The person was sentenced to death, marking a significant moment in the judicial process. The courtroom, filled with tension and gravity, reflects the serious nature of the proceedings. The sketch captures the atmosphere and the weight of the decision rendered by the court.

------------------------------------------------------------------------------------------------

*Rank No.2*, *Cosine Similarity* = 0.4186, *Modality* = text

The image shows a former general, who has been sentenced to life in prison for his role in the murder of a Catholic bishop during Argentina's 1976–83 military dictatorship. The trial revealed documents, including letters from the Vatican archives provided by Pope Francis, which showed the bishop's denunciation of the regime's abuses. The general was found guilty of ordering the murder of Bishop Enrique Angelelli in 1976, marking a significant conviction of a junta-era official for the killing of a high-ranking cleric.

------------------------------------------------------------------------------------------------

*Rank No.3*, *Cosine Similarity* = 0.3994, *Modality* = text

On October 3, 2011, in a courtroom filled with emotional tension, Amanda Knox's father is embraced by his wife following the announcement that Amanda had won her appeal against her murder conviction. The atmosphere is charged with relief and joy as supporters and family members react to the verdict. The image captures a poignant moment of familial support and celebration amidst the wider context of a highly publicized and dramatic legal battle.

------------------------------------------------------------------------------------------------

*Rank No.4*, *Cosine Similarity* = 0.3718, *Modality* = text

Sudheendra Kulkarni was attacked with black ink, leaving his face and head covered. This incident occurred in public, attracting media attention and police presence, as seen in the image. Kulkarni was subsequently taken to a hospital to have the ink removed. The event highlighted tensions and provoked widespread reactions, underscoring the volatile nature of public discourse.

------------------------------------------------------------------------------------------------

*Rank No.5*, *Cosine Similarity* = 0.3698, *Modality* = text

The Rev Sidney Davis leads mourners in a community prayer service at Second Presbyterian Church in Charleston, following the tragic shooting that claimed the lives of nine black worshipers. This gathering reflects the communal grief and solidarity in the face of violence, as mourners join hands in prayer. The event underscores ongoing discussions about race and gun control, issues highlighted during President Obama's presidency. The somber atmosphere is a reminder of the challenges and unresolved issues surrounding racial tensions and gun violence in America.

---

**GR-CLIP Top-5 Results**

*Rank No.1*, *Cosine Similarity* = 0.4265, *Modality* = image (**Ground truth**)

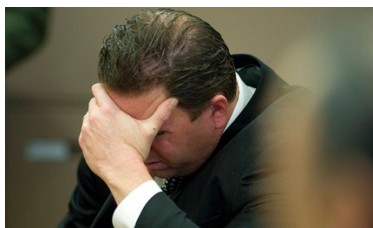

------------------------------------------------------------------------------------------

*Rank No.2*, *Cosine Similarity* = 0.3605, *Modality* = text

In this courtroom sketch, a solemn scene unfolds as the individual is depicted during the sentencing phase of a high-profile trial. The person was sentenced to death, marking a significant moment in the judicial process. The courtroom, filled with tension and gravity, reflects the serious nature of the proceedings. The sketch captures the atmosphere and the weight of the decision rendered by the court.

------------------------------------------------------------------------------------------

*Rank No.3*, *Cosine Similarity* = 0.3365, *Modality* = text

The image shows a former general, who has been sentenced to life in prison for his role in the murder of a Catholic bishop during Argentina's 1976–83 military dictatorship. The trial revealed documents, including letters from the Vatican archives provided by Pope Francis, which showed the bishop's denunciation of the regime's abuses. The general was found guilty of ordering the murder of Bishop Enrique Angelelli in 1976, marking a significant conviction of a junta-era official for the killing of a high-ranking cleric.

------------------------------------------------------------------------------------------

*Rank No.4*, *Cosine Similarity* = 0.3224, *Modality* = multimodal

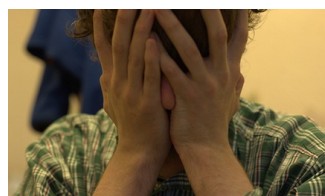 MPs are raising concerns about the lack of access to inpatient mental health services for young people, highlighting cases like Nikki Mattocks, who faced significant delays and inadequate support. Despite her struggles with severe mental health issues, she experienced a fragmented care system, resulting in repeated emergency visits and admissions to distant psychiatric units. This lack of continuity and proximity to family exacerbated her condition. The parliamentary report underscores the urgent need for early intervention and better resource allocation to prevent further harm to vulnerable youths.

------------------------------------------------------------------------------------------

*Rank No.5*, *Cosine Similarity* = 0.2956, *Modality* = text

On October 3, 2011, in a courtroom filled with emotional tension, Amanda Knox's father is embraced by his wife following the announcement that Amanda had won her appeal against her murder conviction. The atmosphere is charged with relief and joy as supporters and family members react to the verdict. The image captures a poignant moment of familial support and celebration amidst the wider context of a highly publicized and dramatic legal battle.

