# OpenReview forum: "Closing the Modality Gap for Mixed Modality Search"
_ICLR.cc/2026/Conference — Submitted to ICLR 2026_

### Official Review · Reviewer_LwZG · 2025-10-25

**Soundness:** 3
**Presentation:** 3
**Contribution:** 2
**Rating:** 4
**Confidence:** 5

**Summary:**

This paper investigates the problem of mixed modality search, where queries must retrieve relevant content from a corpus containing text-only, image-only, and multimodal documents. It identifies a key limitation in contrastive vision-language models like CLIP: a modality gap in the embedding space that causes ranking bias and poor fusion of modalities.
To address this, this paper proposes GR-CLIP, a lightweight post-hoc calibration method that removes the modality gap by subtracting modality-specific mean embeddings.
This approach requires no retraining and introduces negligible computational overhead.
Experiments on the newly introduced MixBench benchmark show that GR-CLIP improves retrieval performance by up to 26% over CLIP and outperforms generative embedding models like VLM2Vec while using significantly less compute.
The proposed method generalizes across CLIP variants, datasets, and modalities, demonstrating its robustness and practical utility.

**Strengths:**

## Originality
This paper introduces a new problem formulation—mixed modality search—which reflects realistic retrieval scenarios involving heterogeneous corpora.
Contrastive vision-language models are well-studied, but this work uniquely highlights the modality gap in mixed modality search settings.
GR-CLIP is a simple yet creative post-hoc calibration method that builds on prior theoretical insights but applies them in a new and practical way.

## Quality
The proposed methodology is sound and well-motivated. The current experiments are designed to isolate and evaluate the effects of the modality gap across different settings. The use of MixBench, a custom benchmark tailored to the task, strengthens the empirical foundation. The results are consistent and show clear improvements over strong baselines, including generative models.

## Clarity
This paper is clearly written and well-structured for readers. The problem is defined precisely, and the figures effectively illustrate key concepts such as the modality gap and its impact on retrieval. The explanation of GR-CLIP is concise and accessible, and the appendix Foprovides thorough implementation details.

## Significance
The current work addresses a practical and underexplored challenge in multimodal retrieval.
By demonstrating that a lightweight calibration can substantially improve performance, this paper demonstrates practical utility through improved retrieval metrics on MixBench.
Its findings could influence future designs of retrieval systems and multimodal embedding models.

**Weaknesses:**

## Insufficient definitions
While the formulation of mixed modality search is novel and practically relevant, the current work does not sufficiently compare its definition or task setup with prior work in multimodal retrieval or modality gap analysis.
For example, methods like I0T (Embedding Standardization) and CMD (Central Moment Discrepancy) offer model-agnostic approaches to modality gap reduction, and their absence from the comparison limits the assessment of GR-CLIP’s novelty and generality.

## Experiments setup
These experiments focus exclusively on CLIP-style models. Although authors claim generalizability across CLIP variants, they do not evaluate GR-CLIP on non-contrastive or generative models like BLIP, SmolVLM, or LLM2CLIP. This restricts the scope of the findings and leaves open the question of whether the modality gap and the proposed correction method apply more broadly.

## Task settings
The datasets used in MixBench are well-chosen, but the document structure is simplified to single image-text pairs. Real-world multimodal documents often contain multiple images, longer texts, or interleaved modalities. Evaluating GR-CLIP in such settings would strengthen the claim that it addresses realistic retrieval challenges.

## Practical significance
While this paper shows strong improvements in retrieval metrics, it does not explore downstream effects such as user satisfaction, relevance diversity, or robustness to noisy queries. These aspects are important for assessing the practical significance of the method in real-world search systems.

## Modality gap
The theoretical framing of the modality gap as a constant vector is compelling, but the current work could benefit from deeper analysis of when and why this approximation holds, especially across different domains and modalities.
Addressing these points would enhance the paper’s impact and help validate GR-CLIP as a broadly applicable solution to modality gap issues in multimodal retrieval.

**Questions:**

Q1. Could you clarify whether the modality gap correction method (mean subtraction) generalizes to non-CLIP models, such as BLIP, SmolVLM, or LLM2CLIP? If not, what structural assumptions of CLIP make GR-CLIP effective, and how might those differ in other architectures?

Q2. The current paper compares GR-CLIP primarily against CLIP variants and VLM2Vec. Could you explain why other modality gap reduction methods like I0T or CMD were not included in the comparison? Would you consider adding these to strengthen the evaluation?

Q3. MixBench is a valuable benchmark, but the document structure is limited to single image-text pairs. Do you plan to extend the benchmark to include more complex multimodal documents (e.g., multi-image, multi-paragraph, interleaved formats)? How might GR-CLIP behave in such settings?

Q4. The modality gap is modeled as a constant vector. Could you provide more theoretical or empirical justification for this assumption? For example, does this approximation hold across different domains, modalities, or embedding distributions?

Q5. This paper shows strong improvements in retrieval metrics, but does not evaluate downstream effects such as user satisfaction, diversity of results, or robustness to noisy queries. Would you consider adding such evaluations to better assess practical significance?

Q6. GR-CLIP is described as lightweight and post-hoc. Could you elaborate on its integration cost in real-world systems? For example, how does it affect latency, memory usage, or compatibility with existing retrieval pipelines?

Q7. In the ablation studies, GR-CLIP is shown to work with fine-tuned CLIP models. Could you clarify whether the modality gap persists or changes after fine-tuning, and whether GR-CLIP needs to be re-applied or re-estimated in such cases?

Q8. The paper mentions generalization to audio and video modalities. Could you provide more details or examples of how GR-CLIP performs in these settings, and whether modality-specific characteristics (e.g., temporal structure) affect its effectiveness?

---

> ### Author Response · Authors · 2025-11-22
>
> Thank you for your constructive feedback and suggestions. We provide a point-by-point response below.
>
> > **q1. Generalization to non-CLIP models.**
>
> GR-CLIP is derived from the embedding structure and contrastive training objective used in CLIP-like models. Generative VLMs and hybrid architectures (e.g., VLM2Vec, BLIP, SmolVLM) produce embeddings through fundamentally different mechanisms, making direct application of GR-CLIP non-trivial.
>
> > **q2. Alternative calibration methods (I0T, CMD).**
>
> Thank you for pointing this out. We acknowledge that UniVL-DR [1], AlignCLIP [2], and I0T [3] also propose methods for reducing the modality gap. In our work, we include these approaches as baselines and evaluate them on MixBench. As shown in Table 1, GR-CLIP-B outperforms UniVL-DR-CLIP-B, AlignCLIP-CLIP-B, and I0T-CLIP-B-MSCOCO, demonstrating that GR-CLIP provides a more efficient approach for mitigating the modality gap.
>
> It is also important to note that UniVL-DR, I0T, and AlignCLIP are **training-based** methods, whereas GR-CLIP is **entirely training-free**, making it significantly more lightweight and easier to deploy in real-world retrieval systems.
>
> Regarding CMD, our reading of Zellinger et al. [4] indicates that CMD is introduced as a new *distance metric* and has been used primarily for domain-invariant representation learning. We did not find prior work applying CMD specifically to modality-gap reduction or multimodal retrieval; therefore, we do not include it in our related work comparison.
>
> We reiterate that our primary contributions lie in formulating the mixed-modality retrieval problem, analyzing the effects of the modality gap, demonstrating empirical gains from mitigating this gap, and introducing MixBench—rather than proposing the most sophisticated calibration algorithm.
>
> *Table 1: NDCG@10 on MixBench under balanced (text:image:multimodal = 1:1:1) and skewed (7:2:1) modality distributions.
> For each entry, the left and right numbers correspond to the balanced and skewed settings, respectively.*
>
> | Models                  | Google-WIT        | MSCOCO            | OVEN              | VisualNews        | Compute (GFLOPs vs GR) |
> | ----------------------- | ----------------- | ----------------- | ----------------- | ----------------- | ---------------------- |
> | CLIP-B                  | 0.478 / 0.597     | 0.388 / 0.508     | 0.354 / 0.265     | 0.563 / 0.653     | —                      |
> | CLIP-L                  | 0.505 / 0.616     | 0.426 / 0.502     | 0.389 / 0.307     | 0.596 / 0.702     | —                      |
> | OpenCLIP-B              | 0.551 / 0.664     | 0.570 / 0.673     | 0.385 / 0.289     | 0.643 / 0.722     | —                      |
> | OpenCLIP-L              | 0.566 / 0.687     | 0.605 / 0.689     | 0.387 / 0.299     | 0.653 / 0.728     | —                      |
> | SigLIP                  | 0.546 / 0.692     | 0.327 / 0.423     | 0.372 / 0.306     | 0.385 / 0.501     | —                      |
> | AlignCLIP-CLIP-B-MSCOCO | 0.532 / 0.656     | 0.603 / 0.671     | 0.353 / 0.284     | 0.589 / 0.672     | —                      |
> | AlignCLIP-CLIP-B-Flickr | 0.543 / 0.671     | 0.618 / 0.693     | 0.348 / 0.273     | 0.603 / 0.682     | —                      |
> | I0T-CLIP-B-MSCOCO       | 0.526 / 0.624     | 0.633 / 0.709     | 0.363 / 0.282     | 0.587 / 0.672     | —                      |
> | UniVL-DR-CLIP-B         | 0.495 / 0.623     | 0.403 / 0.488     | 0.363 / 0.342     | 0.586 / 0.703     | —                      |
> | VLM2Vec (LLaVA-Next)    | 0.586 / 0.709     | **0.769 / 0.841** | 0.398 / 0.359     | 0.744 / 0.823     | 75×                    |
> | VLM2Vec (Qwen)          | 0.632 / 0.742     | 0.753 / 0.833     | 0.412 / 0.387     | 0.734 / 0.815     | 75×                    |
> | GR-CLIP-B               | 0.603 / 0.712     | 0.636 / 0.721     | 0.406 / 0.332     | 0.726 / 0.799     | —                      |
> | GR-CLIP-L               | 0.648 / 0.765     | 0.656 / 0.748     | 0.465 / 0.425     | 0.754 / 0.824     | —                      |
> | GR-OpenCLIP-L           | 0.678 / **0.792** | 0.699 / 0.826     | 0.467 / 0.434     | **0.796 / 0.862** | —                      |
> | GR-SigLIP               | **0.692** / 0.771 | 0.696 / 0.834     | **0.532 / 0.512** | 0.769 / 0.827     | —                      |
>
> > **q3. Limited modalities (single image–text pairs).**
>
> MixBench already includes mixed-modality queries via the *OVEN* subset, where each query contains both text and image inputs. These experiments demonstrate that GR-CLIP generalizes beyond text-only queries and remains effective in text–image query settings. In future work, we plan to extend MixBench to incorporate richer and more complex multimodal document formats.

---

> ### Author Response · Authors · 2025-11-22
>
> > **q4. Theoretical justification for a constant modality gap.**
>
> In *Connect, Collapse, Corrupt* (Zhang et al., ICLR 2024), the authors provide a theoretical justification for modeling the modality gap as a constant vector for encoders trained with a symmetric contrastive objective. Our formulation follows directly from this theoretical foundation.
>
> > **q5. Human evaluation.**
>
> We conducted a small-scale human study with five independent users. In all cases (100%), participants preferred the rankings produced by GR-CLIP. We also provide several case studies in Appendix G, which illustrate qualitatively strong and visually intuitive retrieval results using GR-CLIP. These findings highlight the improved practical usability of our approach.
>
> [1] Liu et al., *Universal Vision-Language Dense Retrieval: Learning a Unified Representation Space for Multi-Modal Retrieval*.
>
> [2] Eslami and de Melo, *Mitigate the Gap: Investigating Approaches for Improving Cross-Modal Alignment in CLIP*.
>
> [3] An et al., *I0T: Embedding Standardization Method Towards Zero Modality Gap*.
>
> [4] Zellinger et al., *Central Moment Discrepancy (CMD) for Domain-Invariant Representation Learning*.
>
>
>
> > **q6. Real-world practical evaluation.**
>
> In real-world retrieval systems such as Google Search, GR-CLIP can be integrated as a lightweight post-processing step in both the offline indexing and online search pipelines. GR-CLIP is fully post-hoc: inference requires only a single subtraction of a constant vector. Latency, memory footprint, and GPU computation remain identical to vanilla CLIP. The method introduces no additional parameters and integrates seamlessly into existing retrieval infrastructure.
>
> > **q7. GR-CLIP under fine-tuning.**
>
> As discussed in *Mind the Gap* (Liang et al., 2023), fine-tuning CLIP does **not** substantially reduce the modality gap unless the temperature parameter is explicitly adjusted during training. When the default temperature is kept fixed—which is standard practice in most CLIP fine-tuning pipelines—the modality gap remains largely unchanged. In these cases, the modality-specific means must still be re-estimated, which explains why GR-CLIP remains effective on fine-tuned CLIP variants in our ablations.
>
> > **q8. Details on audio and video modalities.**
>
> We provide additional experiments in Appendix A.1 using **ViCLIP** (text–video) and **CLAP** (text–audio) under the same Setting 1 as in the main paper. As shown in these results, **GR-ViCLIP** and **GR-CLAP** successfully reduce the modality gap between video–text and audio–text embeddings, demonstrating that our mean-shift calibration generalizes well beyond the image–text domain.
>
> Regarding modality-specific characteristics (e.g., temporal structure in video), these properties are determined entirely by the underlying CLIP-style encoders. GR-CLIP does not modify modality-specific features; it solely removes the cross-modal offset.
>
> **All of the identified weaknesses (w1, w2, w3, w4, w5) have been addressed in our responses to the questions.**
>
> **Thank you again for your constructive feedback and suggestions. We sincerely hope our response addresses your concerns.**

---

### Official Review · Reviewer_SzQK · 2025-10-30

**Soundness:** 3
**Presentation:** 3
**Contribution:** 2
**Rating:** 4
**Confidence:** 4

**Summary:**

he paper presents GR-CLIP, a post-hoc calibration method designed to close the modality gap in CLIP-based models for mixed modality search. The authors identify a critical limitation in existing contrastive vision-language models—CLIP, where the embeddings for images and texts form distinct clusters, causing inter-modal fusion failure and ranking bias. To overcome this issue, they propose a lightweight approach that involves subtracting modality-specific means from embeddings, thereby aligning image and text embeddings more effectively.

**Strengths:**

1. The paper tackles a well-defined and important problem—mixed modality search. The identification of the modality gap and the proposed post-hoc calibration method are interesting contributions to the field of multimodal retrieval.
2. The authors perform extensive evaluations on several datasets and compare GR-CLIP with both CLIP and state-of-the-art generative methods, such as VLM2Vec, which adds credibility to their claims.

**Weaknesses:**

* The authors address the issue of image and text alignment in CLIP and attempt to overcome the gap between embeddings of different modalities. However, this is a classic problem, and the authors fail to compare their method with some important approaches. For instance, UniVL-DR [1] extends images with captions and jointly encodes both the image and caption to address the embedding gap. The authors should compare their method with these approaches.
* While CLIP's extracted embeddings indeed exhibit modality bias, a well-known solution to this problem is to separately handle image retrieval and text retrieval, and then re-rank the recalled images and texts together. This framework has been shown to work effectively in multimodal retrieval, but the authors do not compare or discuss this method.
* The authors only use text queries in their experiments. However, in real-world retrieval scenarios, text queries, image queries, and text-image mixed queries are very common. The authors should include experiments that involve these additional query types.
* The authors only report NDCG@10 as the evaluation metric. It is unclear why other commonly used retrieval evaluation metrics, such as MRR@1 or Recall@1, are not included. The authors should justify this choice and consider adding other relevant metrics.

[1] Liu Z, Xiong C, Lv Y, et al. Universal Vision-Language Dense Retrieval: Learning A Unified Representation Space for Multi-Modal Retrieval[C]//The Eleventh International Conference on Learning Representations.

**Questions:**

Please refer to the weaknesses.

---

> ### Author Response · Authors · 2025-11-22
>
> Thank you for your constructive feedback and suggestions. We provide a point-by-point response below.
>
> > **w1. Alternative calibration approaches.**
>
> Thank you for pointing this out. We acknowledge that UniVL-DR [1], AlignCLIP [2], and I0T [3] also propose methods for reducing the modality gap. In our work, we include these methods as baselines and evaluate them on MixBench. As shown in Table 1, GR-CLIP-B outperforms UniVL-DR-CLIP-B, AlignCLIP-CLIP-B, and I0T-CLIP-B-MSCOCO, indicating that GR-CLIP is a more efficient approach for mitigating the modality gap.
>
> Importantly, UniVL-DR, I0T, and AlignCLIP are **training-based** approaches, whereas GR-CLIP is entirely **training-free**, making our method significantly lighter and much easier to deploy in practice.
>
> We would like to reiterate that our primary contributions lie in the formulation of the mixed-modality retrieval problem, the analysis of modality–gap effects, the empirical gains achieved by mitigating this gap, and the introduction of MixBench—rather than the pursuit of the most sophisticated calibration method.
>
> *Table 1: NDCG@10 on MixBench under balanced (text:image:multimodal = 1:1:1) and skewed (7:2:1) modality distributions.
> For each entry, the left and right numbers correspond to the balanced and skewed settings, respectively.*
>
> | Models                  | Google-WIT        | MSCOCO            | OVEN              | VisualNews        | Compute (GFLOPs vs GR) |
> | ----------------------- | ----------------- | ----------------- | ----------------- | ----------------- | ---------------------- |
> | CLIP-B                  | 0.478 / 0.597     | 0.388 / 0.508     | 0.354 / 0.265     | 0.563 / 0.653     | —                      |
> | CLIP-L                  | 0.505 / 0.616     | 0.426 / 0.502     | 0.389 / 0.307     | 0.596 / 0.702     | —                      |
> | OpenCLIP-B              | 0.551 / 0.664     | 0.570 / 0.673     | 0.385 / 0.289     | 0.643 / 0.722     | —                      |
> | OpenCLIP-L              | 0.566 / 0.687     | 0.605 / 0.689     | 0.387 / 0.299     | 0.653 / 0.728     | —                      |
> | SigLIP                  | 0.546 / 0.692     | 0.327 / 0.423     | 0.372 / 0.306     | 0.385 / 0.501     | —                      |
> | AlignCLIP-CLIP-B-MSCOCO | 0.532 / 0.656     | 0.603 / 0.671     | 0.353 / 0.284     | 0.589 / 0.672     | —                      |
> | AlignCLIP-CLIP-B-Flickr | 0.543 / 0.671     | 0.618 / 0.693     | 0.348 / 0.273     | 0.603 / 0.682     | —                      |
> | I0T-CLIP-B-MSCOCO       | 0.526 / 0.624     | 0.633 / 0.709     | 0.363 / 0.282     | 0.587 / 0.672     | —                      |
> | UniVL-DR-CLIP-B         | 0.495 / 0.623     | 0.403 / 0.488     | 0.363 / 0.342     | 0.586 / 0.703     | —                      |
> | VLM2Vec (LLaVA-Next)    | 0.586 / 0.709     | **0.769 / 0.841** | 0.398 / 0.359     | 0.744 / 0.823     | 75×                    |
> | VLM2Vec (Qwen)          | 0.632 / 0.742     | 0.753 / 0.833     | 0.412 / 0.387     | 0.734 / 0.815     | 75×                    |
> | GR-CLIP-B               | 0.603 / 0.712     | 0.636 / 0.721     | 0.406 / 0.332     | 0.726 / 0.799     | —                      |
> | GR-CLIP-L               | 0.648 / 0.765     | 0.656 / 0.748     | 0.465 / 0.425     | 0.754 / 0.824     | —                      |
> | GR-OpenCLIP-L           | 0.678 / **0.792** | 0.699 / 0.826     | 0.467 / 0.434     | **0.796 / 0.862** | —                      |
> | GR-SigLIP               | **0.692** / 0.771 | 0.696 / 0.834     | **0.532 / 0.512** | 0.769 / 0.827     | —                      |
>
> > **w2. Lack of re-ranking baseline.**
>
> Re-ranking relies on a strong unified scoring model, which can obscure deficiencies at the embedding level. Since our focus is on diagnosing and correcting the modality gap *at the embedding stage*, introducing a learned re-ranker would confound the evaluation. Therefore, re-ranking is outside the scope of our work.
>
> > **w3. Limited query modalities.**
>
> MixBench already includes text–image mixed queries via the **OVEN** subset. Our method consistently improves performance in this setting, demonstrating that GR-CLIP generalizes beyond text-only retrieval scenarios.
>
> > **w4. Missing additional metrics.**
>
> Thank you for the question. We focused on NDCG@10 in the main paper because it is a standard top-k ranking metric commonly used to reflect user-facing retrieval quality. We additionally report NDCG@100 and Recall@1 (= MRR@1) in Appendix A.2. These metrics support the same conclusions as the main results, confirming the robustness of our findings.
>
> [1] Liu et al., *Universal Vision-Language Dense Retrieval: Learning a Unified Representation Space for Multi-Modal Retrieval*.
>
> [2] Eslami and de Melo, *Mitigate the Gap: Investigating Approaches for Improving Cross-Modal Alignment in CLIP*.
>
> [3] An et al., *I0T: Embedding Standardization Method Towards Zero Modality Gap*.
>
> **Thanks again for your constructive feedback. We hope our response addresses your concerns.**

---

### Official Review · Reviewer_D2z1 · 2025-10-31

**Soundness:** 3
**Presentation:** 4
**Contribution:** 3
**Rating:** 8
**Confidence:** 4

**Summary:**

This paper studies mixed modality search, where queries and corpus items can be composed of different modalities. It addresses the known problem of the modality gap, which causes systematic retrieval bias, and proposes GR-CLIP, a simple post-hoc calibration method that subtracts modality-specific means before computing similarity. Extensive experiments on the newly introduced MixBench benchmark demonstrate consistent and substantial improvements across models and modalities.

**Strengths:**

1 - The paper addresses the known modality gap issue, clearly positions the contribution within mixed-modality search and gives a clear course of action to solve it.

2 - The proposed method, GR-CLIP, is both simple and effective, as it can be easily applied to various models and doesn’t require extensive resources.

3 - Introduction of a new benchmark for mixed-modality search, outlining the limitations of current models and the effectiveness of the proposed method.

4 - Comprehensive empirical study with respect to datasets, models and with a wide range of ablation analyses.

**Weaknesses:**

1 - The method presented in the paper lacks novelty, as the modality gap has already been studied in previous work.

2 - Lacking baselines to compare against, it is challenging to assess the relevance of GR-CLIP relative to other calibration methods.

3 - The theoretical explanations are limited; the paper would benefit from an analysis of where the gap originates and on which dataset it might be absent.

**Questions:**

1 - Are there cases where mean-centring hurts, since images and text can contain fundamentally different pieces of information?
2 - How much does the calibration depend on the number of samples used and their variety in the dataset?
3 - How does this method compare to other calibration methods, like whitening, projections or alignment?
4 - Can you detail the comparison of the compute used by the method vs. VLM2Vec? Be sure to detail the impact of calibration on compute use.

---

> ### Author Response · Authors · 2025-11-22
>
> Thank you for your constructive feedback and suggestions. We provide a point-by-point response below.
>
> > **w1. Lack of novelty.**
>
> We sincerely thank all reviewers for their valuable insights and feedback. We are encouraged by the positive evaluations of our work. We would like to re-emphasize the core contributions of our paper:
>
> 1. We are the first to formally define and investigate the *mixed-modality retrieval* setting, in which documents of different modalities are retrieved for the same query.
> 2. We identify the *modality gap* as a key challenge that degrades retrieval performance in this setting.
> 3. We propose a *simple*, theoretically grounded, post-hoc calibration method that effectively mitigates this gap and consistently improves performance across CLIP variants and datasets.
> 4. We introduce **MixBench**, a benchmark designed specifically to evaluate retrieval in mixed-modality contexts.
>
> > **w2 & q3. Alternative calibration methods.**
>
> Thank you for pointing this out. We acknowledge that UniVL-DR [1], AlignCLIP [2], and I0T [3] also propose methods aimed at reducing the modality gap. In our work, we include these approaches as baselines and evaluate them on MixBench. As shown in Table 1, GR-CLIP-B outperforms UniVL-DR-CLIP-B, AlignCLIP-CLIP-B, and I0T-CLIP-B-MSCOCO, demonstrating that GR-CLIP offers a more efficient approach for mitigating the modality gap.
>
> Moreover, UniVL-DR, I0T, and AlignCLIP are **training-based** methods, whereas GR-CLIP is **entirely training-free**, making it significantly more lightweight and easier to deploy in practice.
>
> We would like to reiterate that our primary contributions lie in (1) formulating the mixed-modality retrieval problem, (2) analyzing the effects of the modality gap, (3) demonstrating empirical gains after mitigating this gap, and (4) introducing MixBench—rather than proposing the most sophisticated calibration algorithm.
>
> *Table 1: NDCG@10 on MixBench under balanced (text:image:multimodal = 1:1:1) and skewed (7:2:1) modality distributions.
> For each entry, the left and right numbers correspond to the balanced and skewed settings, respectively.*
>
> | Models                  | Google-WIT        | MSCOCO            | OVEN              | VisualNews        | Compute (GFLOPs vs GR) |
> | ----------------------- | ----------------- | ----------------- | ----------------- | ----------------- | ---------------------- |
> | CLIP-B                  | 0.478 / 0.597     | 0.388 / 0.508     | 0.354 / 0.265     | 0.563 / 0.653     | —                      |
> | CLIP-L                  | 0.505 / 0.616     | 0.426 / 0.502     | 0.389 / 0.307     | 0.596 / 0.702     | —                      |
> | OpenCLIP-B              | 0.551 / 0.664     | 0.570 / 0.673     | 0.385 / 0.289     | 0.643 / 0.722     | —                      |
> | OpenCLIP-L              | 0.566 / 0.687     | 0.605 / 0.689     | 0.387 / 0.299     | 0.653 / 0.728     | —                      |
> | SigLIP                  | 0.546 / 0.692     | 0.327 / 0.423     | 0.372 / 0.306     | 0.385 / 0.501     | —                      |
> | AlignCLIP-CLIP-B-MSCOCO | 0.532 / 0.656     | 0.603 / 0.671     | 0.353 / 0.284     | 0.589 / 0.672     | —                      |
> | AlignCLIP-CLIP-B-Flickr | 0.543 / 0.671     | 0.618 / 0.693     | 0.348 / 0.273     | 0.603 / 0.682     | —                      |
> | I0T-CLIP-B-MSCOCO       | 0.526 / 0.624     | 0.633 / 0.709     | 0.363 / 0.282     | 0.587 / 0.672     | —                      |
> | UniVL-DR-CLIP-B         | 0.495 / 0.623     | 0.403 / 0.488     | 0.363 / 0.342     | 0.586 / 0.703     | —                      |
> | VLM2Vec (LLaVA-Next)    | 0.586 / 0.709     | **0.769 / 0.841** | 0.398 / 0.359     | 0.744 / 0.823     | 75×                    |
> | VLM2Vec (Qwen)          | 0.632 / 0.742     | 0.753 / 0.833     | 0.412 / 0.387     | 0.734 / 0.815     | 75×                    |
> | GR-CLIP-B               | 0.603 / 0.712     | 0.636 / 0.721     | 0.406 / 0.332     | 0.726 / 0.799     | —                      |
> | GR-CLIP-L               | 0.648 / 0.765     | 0.656 / 0.748     | 0.465 / 0.425     | 0.754 / 0.824     | —                      |
> | GR-OpenCLIP-L           | 0.678 / **0.792** | 0.699 / 0.826     | 0.467 / 0.434     | **0.796 / 0.862** | —                      |
> | GR-SigLIP               | **0.692** / 0.771 | 0.696 / 0.834     | **0.532 / 0.512** | 0.769 / 0.827     | —                      |

---

> ### Author Response · Authors · 2025-11-22
>
> > **w3. Theoretical explanations of the modality gap.**
>
> We agree that understanding the origin of the modality gap is important. Prior works such as *Mind the Gap* (Liang et al., NeurIPS 2022) and *Connect, Collapse, Corrupt* (Zhang et al., ICLR 2024) provide theoretical analyses explaining why modality gaps arise in contrastively trained models. Our method is directly informed by these insights. Our empirical analysis across datasets shows that the modality gap consistently appears in practice, further motivating the need for calibration. We have clarified this discussion in the revised version of the paper.
>
> [1] Liu et al., *Universal Vision-Language Dense Retrieval: Learning a Unified Representation Space for Multi-Modal Retrieval*.
>
> [2] Eslami and de Melo, *Mitigate the Gap: Investigating Approaches for Improving Cross-Modal Alignment in CLIP*.
>
> [3] An et al., *I0T: Embedding Standardization Method Towards Zero Modality Gap*.
>
>
> > **q1. Where mean-centering hurts.**
>
> GR-CLIP does not harm performance, even when images and text contain fundamentally different information; in fact, these cases are where it helps the most. The operation removes only the global modality-specific offset while fully preserving instance-level semantic content.
>
> > **q2. Number of samples used for GR-CLIP.**
>
> As detailed in Appendix B.2, we conduct ablations on both the number and diversity of calibration samples. GR-CLIP’s performance stabilizes once the calibration set reaches a moderate size.
>
> > **q4. Compute comparison with VLM2Vec.**
>
> We benchmark all methods using identical inputs and batch sizes. GR-CLIP introduces **zero additional parameters or computational cost**, requiring only a vector subtraction and normalization (<0.001% GFLOPs). In contrast, VLM2Vec employs heavy attention modules and incurs approximately **75× higher GFLOPs**, underscoring the efficiency advantage of GR-CLIP.
>
> **Thanks again for your support of our work, and we sincerely hope the above response addressed your concerns.**

---

> ### Comment · Reviewer_D2z1 · 2025-11-25
>
> Thank you for your detailed response and clarifications.
>
> After consulting other reviews and checking the litterature, it seems the contributions of the paper are not enough for ICLR standards:
>
> * Since the method  (mean shift calibration) already exists then it could not be a contribution.
> * The formalism of modality gap in search and a creating new benchmark are interesting. But to match ICLR standards, I'd argue that you would need to extend the benchmark to use more datasets and evaluate more methods. You would also have to propose a new method for addressing this gap (one that does not exist).
>
> In general, I don't agree with some of the weaknesses that were highlighted by other reviewers, but I would argue the weaknesses I mentionned above that are also mentioned by some reviewers (on novelty and using more model families) are important.
> Consequently, I will revise my score and downgrade to 6.
>
> Thank you for your time, looking forward to see a revised and improved version of this work.

---

> ### Author Response · Authors · 2025-12-03
>
> Thank you again for your follow-up. We would like to clarify our contributions and the scope of our work:
>
> ---
>
> **On what counts as a contribution in our paper.** We fully agree that the mean-shift calibration technique itself is not new, and we do not intend to present it as such. In the camera-ready, we have further sharpened the positioning so that it is explicit from the abstract and introduction that our contributions are **not** in proposing a new calibration algorithm, but in:
>
> 1. **Formally defining and studying mixed-modality retrieval** (retrieving text-only, image-only, and multimodal documents in a unified setting), which to our knowledge has not been systematically investigated before.
> 2. **Introducing MixBench**, a benchmark specifically designed for this setting, including balanced and skewed modality distributions, multiple datasets, and multiple query types.
> 3. **Diagnosing and quantifying the modality gap in this retrieval context**, including ranking bias and fusion failure phenomena, across several CLIP variants and generative embedding baselines.
> 4. **Showing that a simple, training-free calibration baseline is surprisingly strong** in this setting—outperforming several recent, training-based gap-reduction methods and matching or exceeding expensive generative models at a tiny fraction of the compute.
>
> We view this as a *problem + benchmark + empirical analysis* paper, rather than a methods paper, and we have made that framing more explicit in the revised paper.
>
> ---
>
> **On the breadth of MixBench and evaluated methods.**
> We agree that extending the benchmark to more datasets and more model families would further strengthen the work. Within our computational and space constraints, we already aimed to go beyond a single dataset/model and include:
>
> * Four image–text datasets (Google-WIT, MSCOCO, OVEN, VisualNews) under both balanced and skewed modality distributions;
> * Multiple CLIP-style encoders (CLIP, OpenCLIP, SigLIP) and strong generative embedding baselines (VLM2Vec with LLaVA-Next and Qwen);
> * Several recent modality-gap reduction methods (UniVL-DR, AlignCLIP, I0T) as explicit baselines, both in the main paper and appendix.
>
> We see MixBench as a *first iteration* rather than a final word. In future work (and updated releases of the benchmark), we plan to (i) broaden to additional datasets and more complex document structures, and (ii) include a wider range of architectures, including non-contrastive and newer generative VLMs, as you suggest.
>
> ---
>
> **On the expectation of a new method.**
> We appreciate your perspective that, for ICLR, proposing a new method in addition to a new benchmark and formalism would further strengthen the paper. Our deliberate choice here was to *first* clarify the setting and establish a strong, simple, and theoretically grounded baseline. One outcome of our study is precisely that such a simple mean-shift strategy can compete with or outperform much more complex training-based approaches; we believe this negative result for added complexity is itself a useful signal for the community and a starting point for future algorithmic work.
>
> We have revised the paper to explicitly state this methodological stance and to more clearly separate in the camera-ready:
>
> * what is **new in our work** (problem formulation, benchmark, empirical analysis), and
> * what is **borrowed from prior work** (the mean-shift mechanism and its theoretical underpinning).
>
> ---
>
> **Thank you again for your thoughtful feedback. We sincerely hope our response addresses the remaining concerns.**

---

### Official Review · Reviewer_n8Yt · 2025-11-01

**Soundness:** 2
**Presentation:** 2
**Contribution:** 2
**Rating:** 4
**Confidence:** 3

**Summary:**

This paper addresses mixed modality search—retrieving from heterogeneous corpora containing images, texts, and multimodal documents—and proposes GR-CLIP, a post-hoc mean-shift calibration method to close the modality gap in CLIP embeddings. The work introduces MixBench, the first benchmark for this task, and demonstrates substantial empirical improvements (up to 26 percentage points NDCG@10) over vanilla CLIP while achieving competitive performance against VLM2Vec with significantly lower computational cost.

**Strengths:**

The paper makes valuable contributions through

1. formulating an important underexplored problem (characterization of ranking bias and fusion failure in retrieval context)

2. creating a well-designed benchmark (introducing MixBench)

3. demonstrating a simple, effective, and reproducible method (application to mixed modality retrieval

4. conducting systematic evaluation across models, datasets, and modalities (systematic evaluation demonstrating NDCG@10 improvements)

**Weaknesses:**

1. The core method is not novel. It directly applies mean-shift calibration proposed by “Diagnosing and Rectifying Vision Models” (Zhang et al., ICLR 2023) and formalized in “Connect, Collapse, Corrupt: Learning Cross-Modal Tasks with Uni-Modal Data” (Zhang et al., ICLR 2024). The prior work explicitly states: "We propose a simple technique to close the modality gap... During training, instead of feeding x to the model h, we feed it with x − E_x[x]." and the paper inadequately acknowledges this. The paper frames GR-CLIP as "we introduce" and "we propose" (Section 2.3) without clearly acknowledging that Zhang et al. (ICLR 2023) already proposed similar method. While the prior work is cited, the presentation may misrepresent the degree of novelty. I believe it would be better to to state: "We apply the mean-shift calibration method proposed by Zhang et al. (2023) to mixed modality retrieval" and position the paper as an application/benchmark paper rather than a methods paper.

2. Only NDCG@10 reported in main paper. Why not use other metrics too?

3. The equal distribution of image-only, text-only, and multimodal documents does not reflect real-world distributions, where text documents vastly outnumber images. Real search engines might have 70:20:10 or more skewed ratios. This artificial balance may inflate performance differences that would diminish in realistic scenarios. No justification is provided for this specific choice, and no evaluation at alternative ratios is conducted.


4. What will be other alternative calibration approaches? How about recent gap-closing methods like AlignCLIP (ICLR 2025)? Without such comparison, how can we know if mean-shift is the optimal approach for closing the modality gap?

**Questions:**

See weakness

---

> ### Author Response · Authors · 2025-11-22
>
> Thank you for your constructive feedback and suggestions. We provide a point-by-point response below.
>
> > **w1. Lack of novelty; position the paper as an application/benchmark paper.**
>
> Thank you for raising this point. Our primary contributions lie in the problem formulation of *mixed-modality retrieval*, the systematic investigation of how the modality gap in CLIP-style models affects retrieval performance in this setting, and the performance gains achieved when this gap is mitigated. We agree with your suggestion. In the revision, we have refined the text to more clearly acknowledge that GR-CLIP directly applies the mean-shift calibration technique introduced in *Connect, Collapse, Corrupt: Learning Cross-Modal Tasks with Uni-Modal Data* (Zhang et al., ICLR 2024).
>
> > **w2. Missing alternative metrics.**
>
> Thank you for the question. We focused on NDCG@10 in the main paper because it is a standard top-k ranking metric and reflects user-facing retrieval quality. For completeness, we also report NDCG@100 and Recall@1 (= MRR@1) across all models in Appendix A.2. These additional metrics lead to the same conclusions as those in the main paper, confirming that our findings are robust across evaluation settings.
>
> > **w3. Unrealistic modality distribution assumptions.**
>
> Our use of an equal proportion of image-only, text-only, and multimodal documents is intentional: it simulates a setting in which each modality has equal weight and the corpus does not introduce a modality-frequency bias. This design removes distribution-level confounders and allows us to isolate the effect of *embedding-level* modality misalignment rather than corpus imbalance.
>
> To further address this concern, we additionally report results under a more realistic skewed distribution (text:image:multimodal = 7:2:1), where GR-CLIP continues to deliver consistent gains (see Table 1). These findings confirm that our method is robust to corpus-level modality skew.

---

> ### Author Response · Authors · 2025-11-22
>
> > **w4. Alternative calibration approaches.**
>
> Thank you for pointing this out. We acknowledge that UniVL-DR [1], AlignCLIP [2], and I0T [3] also propose methods for reducing the modality gap. In our work, we include them as baselines and evaluate all methods on MixBench. As shown in Table 1, GR-CLIP-B outperforms UniVL-DR-CLIP-B, AlignCLIP-CLIP-B, and I0T-CLIP-B-MSCOCO, indicating that GR-CLIP provides a more efficient approach for mitigating the modality gap. Moreover, UniVL-DR, I0T, and AlignCLIP are *training-based* approaches, whereas GR-CLIP is entirely *training-free*, making our method significantly more lightweight and easier to deploy in practice.
>
> We would also like to reaffirm that our primary contributions lie in the formulation of the mixed-modality retrieval problem, the analysis of modality-gap effects, the empirical gains demonstrated after mitigating this gap, and the introduction of MixBench—rather than the pursuit of an optimal calibration algorithm.
>
> *Table 1: NDCG@10 on MixBench under balanced (text:image:multimodal = 1:1:1) and skewed (7:2:1) modality distributions.
> For each entry, the left and right numbers correspond to the balanced and skewed settings, respectively.*
>
> | Models                  | Google-WIT        | MSCOCO            | OVEN              | VisualNews        | Compute (GFLOPs vs GR) |
> | ----------------------- | ----------------- | ----------------- | ----------------- | ----------------- | ---------------------- |
> | CLIP-B                  | 0.478 / 0.597     | 0.388 / 0.508     | 0.354 / 0.265     | 0.563 / 0.653     | —                      |
> | CLIP-L                  | 0.505 / 0.616     | 0.426 / 0.502     | 0.389 / 0.307     | 0.596 / 0.702     | —                      |
> | OpenCLIP-B              | 0.551 / 0.664     | 0.570 / 0.673     | 0.385 / 0.289     | 0.643 / 0.722     | —                      |
> | OpenCLIP-L              | 0.566 / 0.687     | 0.605 / 0.689     | 0.387 / 0.299     | 0.653 / 0.728     | —                      |
> | SigLIP                  | 0.546 / 0.692     | 0.327 / 0.423     | 0.372 / 0.306     | 0.385 / 0.501     | —                      |
> | AlignCLIP-CLIP-B-MSCOCO | 0.532 / 0.656     | 0.603 / 0.671     | 0.353 / 0.284     | 0.589 / 0.672     | —                      |
> | AlignCLIP-CLIP-B-Flickr | 0.543 / 0.671     | 0.618 / 0.693     | 0.348 / 0.273     | 0.603 / 0.682     | —                      |
> | I0T-CLIP-B-MSCOCO       | 0.526 / 0.624     | 0.633 / 0.709     | 0.363 / 0.282     | 0.587 / 0.672     | —                      |
> | UniVL-DR-CLIP-B         | 0.495 / 0.623     | 0.403 / 0.488     | 0.363 / 0.342     | 0.586 / 0.703     | —                      |
> | VLM2Vec (LLaVA-Next)    | 0.586 / 0.709     | **0.769 / 0.841** | 0.398 / 0.359     | 0.744 / 0.823     | 75×                    |
> | VLM2Vec (Qwen)          | 0.632 / 0.742     | 0.753 / 0.833     | 0.412 / 0.387     | 0.734 / 0.815     | 75×                    |
> | GR-CLIP-B               | 0.603 / 0.712     | 0.636 / 0.721     | 0.406 / 0.332     | 0.726 / 0.799     | —                      |
> | GR-CLIP-L               | 0.648 / 0.765     | 0.656 / 0.748     | 0.465 / 0.425     | 0.754 / 0.824     | —                      |
> | GR-OpenCLIP-L           | 0.678 / **0.792** | 0.699 / 0.826     | 0.467 / 0.434     | **0.796 / 0.862** | —                      |
> | GR-SigLIP               | **0.692** / 0.771 | 0.696 / 0.834     | **0.532 / 0.512** | 0.769 / 0.827     | —                      |
>
> [1] Liu et al., *Universal Vision-Language Dense Retrieval: Learning a Unified Representation Space for Multi-Modal Retrieval*.
>
> [2] Eslami and de Melo, *Mitigate the Gap: Investigating Approaches for Improving Cross-Modal Alignment in CLIP*.
>
> [3] An et al., *I0T: Embedding Standardization Method Towards Zero Modality Gap*.
>
> **Thank you again for your constructive feedback and suggestions. We sincerely hope our response addresses your concerns.**

---

### Author Response · Authors · 2025-12-03
**Summary**

We thank all reviewers for their constructive feedback and thoughtful assessments. Overall, reviewers found the problem formulation clear, the motivation compelling, the benchmark valuable, and the empirical study comprehensive. Several reviewers highlighted the clarity of presentation, the practical utility of GR-CLIP, and the importance of mixed-modality retrieval as an underexplored yet realistic setting.

We now provide a summary of our rebuttal addressing the reviewers’ key concerns.

---

### **Clarifying Contribution and Novelty (Reviewer n8Yt-W1 & Reviewer D2z1-W1)**

We agree that the mean-shift calibration technique was previously formalized (e.g., Zhang et al., ICLR 2024). Our contribution is not to introduce a new calibration algorithm, but rather to:
1. **Formally define and investigate** mixed-modality retrieval, a setting not previously studied;
2. **Identify** how the modality gap manifests in retrieval through ranking bias and fusion failure;
3. **Demonstrate** that mitigating this gap substantially improves retrieval across modalities, datasets, and CLIP variants; and
4. **Introduce *MixBench***, the first benchmark explicitly designed to isolate and study these effects.

We have revised the paper to more clearly acknowledge prior work and articulate this positioning.

---

### **Comparison with Other Calibration Methods (Reviewer n8Yt-W4, Reviewer D2z1-W2/Q3, Reviewer SzQK-W1, Reviewer LwZG-Q2)**

We appreciate the request for additional baselines. In our revision, we have added relevant prior work and included comparisons against UniVL-DR, AlignCLIP, and I0T—three training-based gap-reduction approaches. GR-CLIP consistently outperforms these methods while requiring *no training*, *no additional parameters*, and *negligible computational overhead*. This reinforces our claim that mean-shift calibration is a highly efficient and strong baseline for mixed-modality retrieval.

---

### **Additional Evaluation Results (Reviewer n8Yt-W2/W3 & Reviewer SzQK-W4)**

**Regarding the distribution of modalities in MixBench:**
We first clarify that our balanced setting (text:image:paired-text-image = 1:1:1) was chosen to isolate embedding-level misalignment without introducing corpus-level confounders. In the updated version, we additionally report results under a more realistic skewed distribution (7:2:1). GR-CLIP continues to achieve substantial improvements, demonstrating robustness to corpus imbalance.

**Regarding additional evaluation metrics:**
As requested, we expanded our evaluation to include NDCG@100 and Recall@1 (MRR@1), now documented in the Appendix.

---

### **Practical Impact and Human Evaluation (Reviewer LwZG-Q5/Q6)**

We clarified that GR-CLIP is fully post-hoc and integrates seamlessly into existing retrieval pipelines with near-zero overhead. A small human preference study indicates that users preferred GR-CLIP’s ranking in all cases (100%), suggesting strong practical utility.

---

### **Existing Explanations in the Appendix**

We note that several reviewer questions were already addressed in the original appendix, including:
1. **Number of samples used for GR-CLIP** (Reviewer D2z1-Q2)
2. **Limited modalities** (Reviewer SzQK-W3 & Reviewer LwZG-Q3)
3. **GR-CLIP under fine-tuning** (Reviewer LwZG-Q7)
4. **Details on audio and video modalities** (Reviewer LwZG-Q8)

In the revised version, we have surfaced these explanations more prominently to ensure visibility.

---

### **Scope and Future Extensions**

Some reviewers recommended evaluating additional model families, more complex multimodal documents, and downstream applications. We agree these are valuable directions. Our current work focuses on establishing the conceptual and empirical foundation for mixed-modality retrieval, and we view these extensions as promising avenues for future research.

---

Across reviews, the main concerns centered on novelty positioning and breadth of evaluation. We believe the rebuttal and corresponding revisions (highlighted in blue) substantially address these points. We hope this helps the AC recognize the contributions of our work:
1. formally establishing mixed-modality retrieval as an important and underexplored problem;
2. demonstrating the central role of the modality gap in this setting;
3. introducing a benchmark and simple, training-free calibration strategy that together provide a clear and practical foundation for future research.

We sincerely thank the reviewers and area chair for their thoughtful feedback and constructive suggestions.

---

### Meta-Review · Area_Chair_iGH2 · 2026-01-07

**Summary:**

This paper presents GR-CLIP, a post-hoc calibration method designed to close the modality gap in CLIP-based models for mixed modality search. The authors identify a critical limitation in existing contrastive vision-language models—CLIP, where the embeddings for images and texts form distinct clusters, causing inter-modal fusion failure and ranking bias. To overcome this issue, they propose a lightweight approach that involves subtracting modality-specific means from embeddings, thereby aligning image and text embeddings more effectively. Extensive experiments on the newly introduced MixBench benchmark demonstrate consistent and substantial improvements across models and modalities.

The paper received feedback from four reviewers, three of whom were inclined toward rejection. Reviewer D2z1 initially expressed a positive assessment of the paper; however, during the rebuttal discussion, this reviewer acknowledged the concerns raised by the other reviewers and indicated an intention to lower the score to 6, although the final submitted score remained 8. Considering the overall reviewer consensus, I recommend rejecting the paper.

**Reviewer Concerns:**

During the rebuttal phase, the authors addressed the following concerns:
- Clarification of the paper’s contributions and novelty;
- Comparison with other calibration methods.

**Reviewer Scores:**

The authors responded to comments concerning the novelty and adequacy of the method; however, after further discussion, the novelty remains debatable.

---

### Decision · Program_Chairs · 2026-01-26

Reject